# DPad: Efficient Diffusion Language Models with Suffix Dropout

**Xinhua Chen**,[*] **Sitao Huang**,[*] **Cong Guo**[*†], **Chiyue Wei, Yintao He, Jianyi Zhang,**
**Hai "Helen" Li & Yiran Chen**
Duke University

## Abstract

Diffusion-based Large Language Models (dLLMs) parallelize text generation by framing decoding as a denoising process, but suffer from high computational overhead since they predict all future suffix tokens at each step while retaining only a small fraction. We propose **Diffusion Scratchpad** (*DPad*), a training-free method that restricts attention to a structured subset of suffix tokens, preserving fidelity while eliminating redundancy. *DPad* integrates two strategies: (i) a *sliding window*, which maintains a fixed-length suffix window, and (ii) *distance-decay dropout*, which deterministically removes distant suffix tokens before attention computation. This concise design is compatible with existing optimizations such as parallel decoding and prefix caching, and lends itself to a lightweight implementation. Comprehensive evaluations across multiple benchmarks on `LLaDA` and `Dream` models demonstrate that *DPad* delivers up to $61.4\times$ speedup over vanilla dLLMs while maintaining comparable accuracy, highlighting its potential for efficient and scalable long-sequence inference. Our open-source implementation is available at `https://github.com/Crys-Chen/DPad.git`.

## 1 Introduction

Large Language Models (LLMs) have become foundational in numerous applications (Vaswani et al., 2017; Devlin et al., 2019; Brown et al., 2020; Ouyang et al., 2022), yet their deployment is often hindered by inference latency. As shown in Fig. 1 (a), the predominant autoregressive framework generates text one token at a time (Radford, 2018; Radford et al., 2019), imposing a sequential constraint that limits speed and scalability (Gu et al., 2018). This has driven interest toward parallel decoding strategies.

Diffusion-based Large Language Models (Li et al., 2022; Austin et al., 2021a; Lou et al., 2024; Shi et al., 2025; Israel et al., 2025) (dLLMs) offer a promising alternative by eliminating sequential dependencies. Formulating text generation as a parallel denoising process, they can predict entire sequences or generate text block-wise (i.e., semi-autoregressively) (Nie et al.; Ye et al., 2025), as in Fig. 1 (b). However, this parallelism often incurs high computational cost (Nie et al., 2025): at each step, predictions for *all* future (suffix) tokens are computed, though only a small fraction are retained. Consequently, although dLLMs can generate multiple tokens in parallel, the resulting throughput gains are undermined by a disproportionate increase in computation, posing a key bottleneck to their widespread adoption (Song et al., 2025b).

To further understand this inefficiency, we analyze the role of suffix tokens under the block-wise masking mechanism in dLLMs and reveal that suffix tokens act as a non-semantic *information reservoir*, or scratchpad. While this scratchpad collects contextual signals across Transformer layers to guide the generation of the current block, it is highly inefficient. We find that most suffix tokens are redundant and low-entropy (see Appx. A), a problem that worsens with distance as their attention scores decay sharply. This redundancy not only creates significant computational overhead but can also degrade generation fidelity.

---

[*]Equal contribution.
[†]Corresponding author: cong.guo@duke.edu

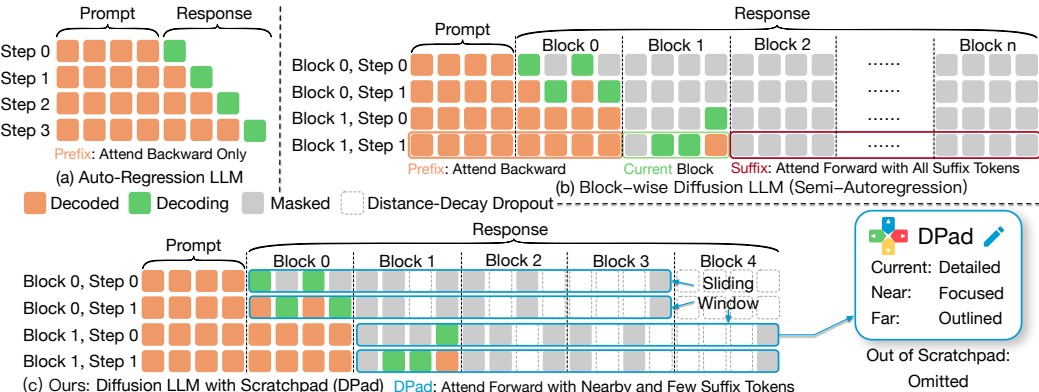

Figure 1: Comparison of (a) autoregressive LLMs, (b) block-wise diffusion LLMs, and (c) our *DPad*. *DPad* restricts suffix attention via: (*i*) Sliding Window: fixed-length suffix window; (*ii*) Distance-decay Dropout: removes distant suffix tokens without computing attention scores.

Based on the above insights, we propose the **Diffusion Scratchpad** (*DPad*), which in the forward direction attends only to a small number of near-suffix tokens, as in Fig. 1 (c). It uses two suffix drop strategies: *sliding window* and *distance-decay dropout*. For the ***sliding window***, inspired by prefix KV-cache optimizations (Beltagy et al., 2020; Xiao et al., 2023), we extend the idea to the suffix. Here, the suffix window has a fixed length and moves forward along with the current block, retaining only a limited number of nearby suffix tokens. In contrast to vanilla dLLMs, where suffix computation increases with the generation sequence length, our design keeps the cost bounded and significantly reduces suffix-related computation.

For the ***distance-decay dropout***, suffix tokens are removed according to their distance from the current block: the farther they are, the higher the dropout ratio, until all tokens beyond the window are omitted. Unlike conventional attention-score pruning (Wang et al., 2021; Kim et al., 2022; Song et al., 2025a), which first computes attention scores and then prunes based on their magnitude, *DPad* predetermines a distance-decay sparse pattern for suffix tokens *prior to* model execution and eliminates them at the very beginning of each step. This further suggests that sparsity is an inherent property of suffix attention. Additionally, the method is extremely simple to deploy, requiring only a few lines of code to implement.

To illustrate the intuition behind *DPad*, we liken it to a real scratchpad used when writing a book, as shown in Fig. 1 (c, right). For the current chapter (i.e., block), the writer (dLLM) devotes the most attention, revising it multiple times, akin to denoising within a diffusion block. The upcoming chapter receives focused drafts for consistency, while much later chapters contain only brief outlines. Naturally, the "writer" should not, and indeed must not, fill all future chapters (all suffix tokens) with low-entropy, repetitive content merely to satisfy the fixed sequence length constraints of current dLLMs. Such uncontrolled filling distracts the "writer's" attention and wastes storage and computation, making it neither sustainable nor scalable.

Finally, we emphasize that *DPad* is a ***training-free*** inference strategy that enables efficient, compact generation, overcoming the quality degradation from fixed sequence lengths in conventional dLLMs (Ye et al., 2025; Nie et al.). It delivers robust acceleration across all settings, achieving up to a $10.32\times$ speedup on short sequences. As *DPad* is compatible with existing optimizations (Wu et al.; Liu et al.; Wei et al.), including prefix caching (Wu et al.) and dLLM-Cache (Liu et al.), its benefits compound for long-sequence generation. When combined with these methods, *DPad* achieves up to a $61.39\times$ speedup on `LLaDA-1.5`/GSM8K (1024, 1-shot) while maintaining accuracy, highlighting its potential to unlock new frontiers in dLLM efficiency.

In summary, our contributions are:

- We identify and formalize the *Scratchpad* mechanism in dLLMs, demonstrating that suffix tokens act as a dynamic, cross-layer reservoir (which we term an *Attention Connection*) that guides the denoising process.

- Through systematic empirical analysis, we are the first to reveal three key properties of suffix tokens: inherent sparsity, a distance-decay pattern, and position insensitivity, which expose a major yet overlooked inefficiency in dLLMs.

- We introduce the *Diffusion Lottery Tickets* (DLT) hypothesis for dLLM inference, positing that a sparse set of "winning" suffix tokens suffices for high-quality generation, and we frame *DPad* as a training-free method to discover such tickets on the fly.

- We propose *DPad*, a training-free method that applies distance-decay dropout to suffix tokens. This orthogonal design eliminates substantial redundancy and compounds with existing optimizations, achieving up to $\mathbf{61.39\times}$ speedup while preserving accuracy.

## 2 PRELIMINARY

### 2.1 FOUNDATIONAL PRINCIPLES OF DIFFUSION LARGE LANGUAGE MODELS (DLLMS)

For a generated sequence $\mathbf{x} = (x_1, \ldots, x_L)$, dLLMs leverage a non-autoregressive process (Austin et al., 2021a; Shi et al., 2025; Lou et al., 2024). During training, the model learns to denoise a sample where tokens are masked. This involves two processes: (i) **Forward Masking Process**: This process systematically replaces a proportion of tokens in a clean text sequence $\mathbf{x}_0$ with a special [MASK] token (Nie et al., 2025; Nie et al.), similar to applying noise in a conventional diffusion model (Ho et al., 2020). The replacement is governed by a masking schedule where $t \in [0, 1]$ represents the masked level, and individual tokens in the clean sequence, $x_0^i$ are masked independently with a probability of $t$. (ii) **Reverse Unmasking Process**: The model $p_\theta$ is trained to predict the original tokens given the partially masked sequence $\mathbf{x}_t$, thereby learning to approximate the true unmasking probability $q(\mathbf{x}_0|\mathbf{x}_t)$ (Austin et al., 2021a; Lou et al., 2024). This is achieved by minimizing the negative log-likelihood over the masked tokens (Shi et al., 2025; Ouyang et al., 2022):

$$\mathcal{L}(\theta) = -\mathbb{E}_{t,\mathbf{x}_0,\mathbf{x}_t} \left[ \frac{1}{t} \sum_{i:x_t^i=[\text{MASK}]} \log p_\theta(x_0^i|\mathbf{x}_t) \right] \tag{1}$$

### 2.2 INFERENCE AND SAMPLING IN DLLM

The inference process begins by initializing a sequence $y_0$ through the concatenation of a prompt sequence $\mathbf{r}$ and $L$ mask tokens, denoted as $\mathbf{y}_0 = r \circ ([\text{MASK}])_{i=1}^L$. Let $M_s$ be the set of indices corresponding to masked tokens at step $s$; initially, $M_0 = \{|r| + 1, \ldots, |r| + L\}$. The model then iteratively refines this sequence over steps $s = 1, 2, \ldots, S$ until $M_S = \emptyset$. At each intermediate step $s$, the model $p_\theta$ computes a probability distribution $p_\theta(y_i|y_{s-1})$ for every masked position $i \in M_{s-1}$. From these distributions, the most likely token predictions $\hat{y}_i$ from vocabulary $V$ and their corresponding confidence scores $c_i$ are determined:

$$\hat{y}_i = \underset{v \in V}{\text{argmax}}\, p_\theta(y_i = v|\mathbf{y_{s-1}}) \quad \text{and} \quad c_i = p_\theta(y_i = \hat{y}_i|\mathbf{y_{s-1}}). \tag{2}$$

A scheduling function $\mathcal{G}(s, S)$ determines the number of tokens, $k_s$, to unmask. The set of indices to update, $U_s$, is chosen by selecting the $k_s$ positions from $M_{s-1}$ with the highest confidence scores. The new sequence $\mathbf{y}_s$ is then formed by updating these selected masked position.

### 2.3 RELATED WORK ON DLLM ACCELERATION

Research in dLLMs has moved from fixed Top-k decoding to dynamic, confidence-aware unmasking, that greedily unmask all tokens above a confidence threshold, to reduce generation bottlenecks (Wu et al.; Wei et al.). A parallel focus is cache management to overcome the inapplicability of conventional KV caching for bi-directional attention. Optimizations include reusing the cache for tokens (e.g., prefixes) by observing their key and values ( (Wu et al.; Liu et al.) are stable across inference step, and dynamically evicting entries based on low attention score to further improve efficiency (Song et al., 2025a).

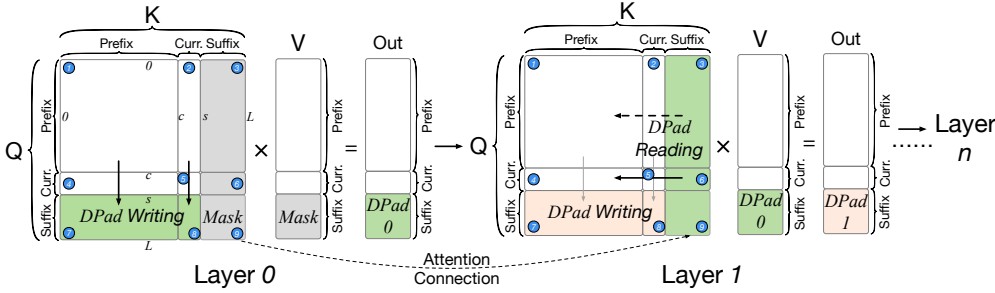

Figure 2: Attention score maps illustrating the **Scratchpad** mechanism in dLLMs. The attention matrix is divided into $3 \times 3$ blocks over *prefix*, *current*, and *suffix*. Blocks 7 and 8 collect information from the prefix and current into the suffix at layer $n$, while Block 6 feeds this stored information back to the current block at layer $(n+1)$.

## 3 METHOD

In this section, we begin by formalizing the *Scratchpad* mechanism in dLLMs, which underpins the role of suffix tokens as an information reservoir. We then introduce *Suffix Dropout*, which combines a fixed-length sliding window that moves with the current block to bound the number of attended suffix tokens, with a distance-decay scheme that progressively prunes more distant ones. Finally, we propose the *Diffusion Lottery Tickets* (DLT) hypothesis, which provides a principled lens for interpreting dropout in dLLMs.

### 3.1 SCRATCHPAD MECHANISM

We first revisit the role of suffix tokens in dLLMs. Due to the masking structure, suffix tokens carry no direct semantic information but instead serve as an *information reservoir*, aggregating signals propagated from prefix tokens across multiple Transformer layers. This latent memory, which we refer to as **DPad** (Diffusion Scratchpad), stabilizes the denoising process by providing contextual support for the current block.

As illustrated in Fig. 2 and the attention score maps, the token sequence can be partitioned into three contiguous segments: Prefix indices $[0, c-1]$, current indices $[c, s-1]$, and suffix indices $[s, L-1]$. The corresponding attention matrix is thus divided into $3 \times 3$ blocks. Among them, Blocks 6, 7, and 8 together define the scratchpad mechanism.

Considering only one head, at layer $n$, queries from prefix and current tokens attend to keys from the suffix region. Formally, the global attention scores are defined as

$$A^{(n)} = \text{Softmax}\left( \frac{Q^{(n)}(K^{(n)})^\top}{\sqrt{d_k}} \right) \in \mathbb{R}^{L \times L}. \tag{3}$$

We can partition $A^{(n)}$ into submatrices corresponding to prefix ($P = [0, c-1]$), current ($C = [c, s-1]$), and suffix ($S = [s, L-1]$). In particular,

$$A_{S,P}^{(n)} = A^{(n)}[s : L, \, 0 : c], \qquad A_{S,C}^{(n)} = A^{(n)}[s : L, \, c : s], \tag{4}$$

represent the attention scores from suffix queries to prefix and current keys, respectively (Blocks 7 and 8 in Fig. 2). Multiplying these attention scores with the value matrix yields the actual outputs at suffix positions:

$$H_S^{(n)} = A_{S,P}^{(n)} V_P^{(n)} \; + \; A_{S,C}^{(n)} V_C^{(n)} \; + \; A_{S,S}^{(n)} V_S^{(n)}. \tag{5}$$

Here, $H_S^{(n)}$ denotes the hidden representations of suffix tokens after attention. This equation shows that suffix tokens integrate information from prefix and current regions, effectively serving as a *DPad-n* that records contextual information. After this aggregation, the outputs $H^{(n)}$ are processed by the subsequent linear transformations (e.g., feed-forward layers and residual connections), which operate independently on each token. At layer $(n+1)$, this stored information can be retrieved by the current block through

$$A_{C,S}^{(n+1)} = A^{(n+1)}[c : s, \, s : L], \tag{6}$$

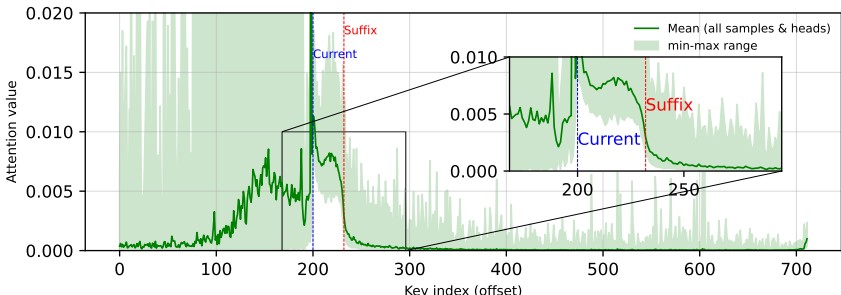

Figure 3: Analysis of the suffix dropout strategy at the final layer 31 using LLaDA-1.5 on GSM8K with max length 512. We gather attention scores from 100 samples across all heads, focusing on current block queries ($A[c{:}s, c\text{-}200{:}]$). The plot shows the mean attention over key indices (green) with min–max range (shaded).

which corresponds to current-to-suffix attention (Block 6 in Fig. 2). This path enables the current block to reuse the information collected in the suffix at the previous layer. In practice, the influence of the suffix on the prefix (Block 3) is negligible (Wu et al.). Therefore, the essential interaction of the scratchpad mechanism lies in the current-to-suffix direction (Block 6), where the suffix serves as temporary memory to assist the ongoing denoising process.

Therefore, we conjecture that the role of suffix tokens resembles a *residual connection* (He et al., 2015), but specialized for attention, which we term an **Attention Connection**. Rather than directly propagating representations, the model compresses high-dimensional signals from both the prefix and the current tokens into the suffix, which is then re-injected into the current block at the subsequent layer. In this way, the suffix serves as a scratchpad that compacts contextual information and forwards it for cross-layer reuse in dLLMs. We provide more analysis about this mechanism in Appx. A.1

### 3.2 SUFFIX DROPOUT STRATEGIES

As discussed in Sec. 1, the scratchpad intuition does not require populating all suffix tokens, which is neither sustainable nor efficient. Empirically, we observe substantial redundancy among suffix tokens (Appx. A), motivating a structured dropout design. As an example, at the final layer 31, Fig. 3 shows a clear distance-dependent decay in attention: nearby tokens dominate, consistent with general attention patterns and prior work on prefix sliding windows (Xiao et al., 2023). We provide additional layer-wise attention visualizations in Appx. A.4.

Based on these findings, we propose two complementary suffix drop strategies, as illustrated in Fig. 1(c):

- (i) **Sliding window** with fixed-length to retain only a bounded number of near-suffix tokens;
- (ii) **Distance-decay dropout** that progressively prunes distant suffix tokens.

Both mechanisms are efficiently realized through a **Gaussian sampling** process, which simultaneously enforces a bounded window and distance-dependent decay.

Formally, for a suffix token at distance $d$ from the suffix boundary, its retention probability $P(d)$ is defined by the right half of a standard normal distribution with effective window size $W$:

$$P(d) = a \cdot \frac{1}{\sigma\sqrt{2\pi}} \exp\left[-\tfrac{1}{2}\left(\frac{\frac{k\sigma}{W}\cdot d - \mu}{\sigma}\right)^2\right], \quad 0 < d \le W, \tag{7}$$

where $\mu = 0$, $\sigma = 1$, and the suffix boundary is the center of the Gaussian distribution. Two hyperparameters, $k$ and $a$, are introduced to control the distribution along the $x$- and $y$-axes: (1) $\frac{k\sigma}{W}$ maps the window size $W$ to $k\sigma$ (e.g., for $W = 256$, setting $k = 3$ ensures $d = 256$ corresponds to $3\sigma$); (2) $a$ scales the overall sampling magnitude vertically. This formulation ensures that tokens farther from the boundary are retained with exponentially decreasing probability, implemented via

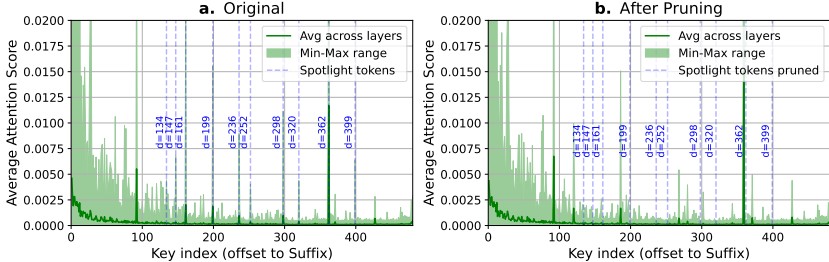

Figure 4: (a) Attention scores of suffix tokens paid by current block tokens ($A[c{:}s, s{:}]$) across layers in LLaDA-1.5, showing overall decay with occasional spikes (e.g., d = 199, 298, 362). (b) After forcibly pruning these spike positions, attention shifts to nearby tokens, indicating that adjacent positions can absorb suffix information (e.g., pruning token 362 shifts the spike to token 359).

rejection sampling. More implementation details and a hyperparameter analysis are provided in Appx. B and Appx. D, respectively.

As a result, suffix attention only needs to focus on nearby tokens, making the suffix dropout window decoupled from the overall sequence length. Unlike the vanilla setting, where the suffix grows with the generation sequence, our approach keeps it constant. This yields a clear computational benefit: suffix dropout effectively reduces suffix-related complexity by one dimension.

### 3.3 DIFFUSION LOTTERY TICKETS HYPOTHESIS

The analysis in Fig. 3 not only reveals the overall decay of current-to-suffix attention, but also occasional sharp *spikes* in the maximum values. These spikes cannot be predicted in advance and, in principle, pruning them could affect model accuracy. To investigate this, we conduct an additional experiment shown in Fig. 4. We first run dLLM inference for one step, then forcibly prune the top 10 highest-attention suffix tokens ("spotlight" tokens) occurring beyond the first 128 positions (e.g., at distances 199, 298, and 362). Surprisingly, pruning such distant suffix tokens, even those corresponding to large spikes, has little effect on model accuracy.

When these spikes are removed, the model shifts its attention to nearby suffix tokens, for example, the spotlight token at **362** in Fig. 4 (a) is replaced by increased attention at its neighbor token **359** in Fig. 4 (b), which effectively *absorbs* the lost information. This behavior is consistent with the strong generalization ability of diffusion models and the fact that suffix tokens are initialized without semantic content: through *DPad* mechanism, suffix tokens can dynamically learn and store information during inference. Consequently, information carried by distant suffix tokens appears largely local *position-adaptive*, and pruning spike positions has almost no impact on final accuracy, as further confirmed in our evaluation experiments, as shown in Table 1.

These observations resonate with the *Lottery Ticket Hypothesis (LTH)* (Frankle & Carbin, 2019), which posits that properly pruned sub-networks with their original initialization can match the performance of dense networks after training. We extend this intuition to dLLMs and propose the **Diffusion Lottery Tickets** (DLT) hypothesis: During inference, the suffix region contains redundant tokens, yet a sparse subset is sufficient to preserve semantic consistency and generation quality. Through *DPad* mech-

Table 1: Accuracy Score of LLaDA-1.5 (Length = 512).

| Method | GSM8K | | HumanEval |
| --- | --- | --- | --- |
| | Strict | Flex. | Score |
| Baseline | 40.5 | **78.7** | 37.8 |
| Top-10 Pruning | **41.7** | 78.6 | **39.0** |

anism, this subset can be adaptively reorganized into "winning tickets" within the forward pass. In this view, suffix dropout becomes a *training-free lottery ticket search*, where Gaussian sampling selects a compact set of suffix tokens that carry the essential information for denoising in dLLMs.

This explains why suffix dropout can be applied *a priori*, without computing exact attention scores, and why it fundamentally differs from prefix cache pruning (Song et al., 2025a; Wang et al., 2021): Prefix tokens carry dense, position-bound semantic information and thus cannot be arbitrarily discarded, whereas suffix tokens act as a flexible, low-rank memory buffer.

Table 2: Consolidated performance of LLaDA-Instruct and Dream-Base on four benchmarks.

| Benchmark | Method | LLaDA-Instruct Efficiency Latency(s)↓ | | TPS↑ | | $\bar{\ell}/\ell_{max}$ | Accuracy (%) Flexible↑ | Strict↑ | Dream-Base Efficiency Latency(s)↓ | | TPS↑ | | $\bar{\ell}/\ell_{max}$ | Accuracy (%) Flexible↑ | Strict↑ |
|---|---|---|---|---|---|---|---|---|---|---|---|---|---|---|---|
| GSM8K *4-shot* | Vanilla | 27.48 | 1.00× | 8.44 | 1.00× | 232 / 256 | 78.39 | 37.38 | 22.30 | 1.00× | 11.43 | 1.00× | 255 / 256 | 75.06 | 74.37 |
| | *+DPad* | 18.35 | 1.50× | 8.76 | 1.04× | 161 / 256 | 78.54 | 63.84 | 10.27 | 2.17× | 12.75 | 1.11× | 131 / 256 | 75.28 | 75.06 |
| | +Par. | 8.55 | 3.21× | 27.14 | 3.22× | 232 / 256 | 78.54 | 38.67 | 13.84 | 1.61× | 18.43 | 1.61× | 255 / 256 | 75.51 | 74.83 |
| | *+Par.+DPad* | **6.64** | **4.14×** | 24.25 | 2.87× | 161 / 256 | 79.76 | 64.97 | **5.24** | **4.25×** | 24.17 | 2.11× | 127 / 256 | 74.83 | 74.75 |
| MATH *4-shot* | Vanilla | 25.40 | 1.00× | 9.79 | 1.00× | 249 / 256 | 33.58 | 8.42 | 21.01 | 1.00× | 12.19 | 1.00× | 256 / 256 | 34.06 | 37.76 |
| | *+DPad* | 21.61 | 1.18× | 9.75 | 1.00× | 211 / 256 | 33.42 | 28.04 | 16.64 | 1.26× | 15.33 | 1.26× | 255 / 256 | 34.14 | 37.64 |
| | +Par. | 9.91 | 2.56× | 25.09 | 2.56× | 249 / 256 | 33.40 | 8.76 | 8.82 | 2.38× | 29.03 | 2.38× | 256 / 256 | 35.12 | 38.62 |
| | *+Par.+DPad* | **9.20** | **2.76×** | 22.93 | 2.34× | 211 / 256 | 33.30 | 27.98 | **7.72** | **2.72×** | 33.04 | 2.71× | 255 / 256 | 34.44 | 38.32 |
| HumanEval *0-shot* | Vanilla | 34.67 | 1.00× | 13.64 | 1.00× | 473 / 512 | 43.90 | – | 28.49 | 1.00× | 17.93 | 1.00× | 511 / 512 | 51.22 | – |
| | *+DPad* | 27.41 | 1.26× | 15.96 | 1.17× | 438 / 512 | 47.56 | – | 8.20 | 3.47× | 26.83 | 1.50× | 220 / 512 | 51.22 | – |
| | +Par. | 11.48 | 3.02× | 41.40 | 3.04× | 475 / 512 | 43.29 | – | 14.15 | 2.01× | 36.11 | 2.01× | 511 / 512 | 53.05 | – |
| | *+Par.+DPad* | **9.14** | **3.79×** | 47.86 | 3.51× | 438 / 512 | 46.34 | – | **4.06** | **7.01×** | 52.62 | 2.93× | 214 / 512 | 52.44 | – |
| MBPP *3-shot* | Vanilla | 62.11 | 1.00× | 4.82 | 1.00× | 299 / 512 | 15.00 | – | 49.15 | 1.00× | 10.42 | 1.00× | 512 / 512 | 52.40 | – |
| | *+DPad* | 15.89 | 3.91× | 6.85 | 1.42× | 109 / 512 | 40.40 | – | 41.36 | 1.19× | 12.38 | 1.19× | 512 / 512 | 52.60 | – |
| | +Par. | 14.26 | 4.36× | 20.99 | 4.36× | 299 / 512 | 15.00 | – | 12.38 | 3.97× | 41.36 | 3.97× | 512 / 512 | 55.40 | – |
| | *+Par.+DPad* | **6.02** | **10.32×** | 18.28 | 3.79× | 110 / 512 | 39.40 | – | **9.86** | **4.98×** | 51.92 | 4.98× | 512 / 512 | 54.80 | – |

## 4 EXPERIMENTS

### 4.1 EXPERIMENTAL SETUP

**Models and Baselines.** All experiments are conducted on an NVIDIA A100 80GB GPU. We evaluate *DPad* on a suite of representative open-source dLLMs: two variants of LLaDA (8B-Instruct and 1.5) (Nie et al.; Zhu et al., 2025) and Dream-v0-Base-7B (Ye et al., 2025). We compare our method against three baselines: the unmodified LLaDA (Nie et al.; Zhu et al., 2025) and Dream (Ye et al., 2025) backbones, denoted as `Vanilla`; the vanilla backbone augmented with parallel decoding (Wu et al.), denoted as `+Par.`; and a version implementing prefix caching (Wu et al.), denoted as `PC.`. Since our dropout already minimizes the computational cost of the suffix, we do not adopt the Dual Cache mechanism proposed in (Wu et al.).

The *DPad* strategy introduces three hyperparameters: a decay rate factor $k$, a magnitude scalar $a$, and a sliding window size. We tuned these on small subsets of each benchmark (see Sec. 4.3.2).

Unless otherwise specified, all experiments use a block size of 32, a batch size of 1, and a confidence threshold of 0.9 for parallel decoding.

**Benchmarks and metrics.** We evaluate on reasoning benchmarks (GSM8K (Cobbe et al., 2021), MATH (Hendrycks et al., 2021)) and code generation benchmarks (HumanEval (Chen et al., 2021), MBPP (Austin et al., 2021b)). **Accuracy** is reported with task-specific metrics (e.g., pass@1, flexible/strict-match). **Efficiency** is reported as (a) mean end-to-end latency per sample and (b) tokens-per-second (TPS). We also report the average generated length $\bar{\ell}$ relative to the maximum allowed sequence length $\ell_{max}$ as $\bar{\ell}/\ell_{max}$ to make length effects explicit.

### 4.2 MAIN RESULTS

Across all four benchmarks, *DPad* improves latency and achieves comparable or higher TPS relative to `Vanilla`, while preserving or improving accuracy, as shown in Tbl. 2. When combined with parallel decoding (`+Par.`), *DPad* achieves additional improvement beyond `+Par.` alone; results for LLaDA-1.5 are in Appx. C.1.

**Efficiency.** *DPad* contributes to latency gains through two mechanisms: (1) *reduced suffix complexity*, which shifts suffix computation from quadratic toward linear scaling; (2) *more concise generations*, as evidenced by the $\bar{\ell}/\ell_{max}$ metric. This conciseness is beneficial, as case studies in Appx. E demonstrate that suffix dropout suppresses the generation of tokens that play an auxiliary role and do not contribute significantly to reasoning.

Overall, focusing on latency as the primary efficiency metric, *DPad* yields 1.18×–3.91× speedups over `Vanilla`. With parallel decoding, total speedups reach 2.72×–10.32× relative to `Vanilla`. This complementarity arises because the two methods target orthogonal bottlenecks: *DPad* elimi-

nates redundant KV-token computation, while parallel decoding mitigates dependency constraints. By combining these approaches, we exploit both finer-grained token pruning and safe multi-token prediction, yielding substantial efficiency gains.

In our analysis, we observe a subtle distinction between latency- and TPS-based efficiency metrics. While *DPad* consistently reduces per-sample latency, its TPS gains may appear less pronounced because it often encourages the model to generate more concise and complete responses, reaching the end-of-sequence token earlier. As discussed in Appx. E, this behavior reflects not a limitation but an improvement: the model terminates naturally rather than exhausting its context with low-quality continuations. However, when the generation length is substantially reduced, lower GPU utilization can mechanically decrease TPS. Since the extra tokens produced by baselines often carry low semantic value, we report raw throughput without post-processing. This also highlights a broader point: the community may need to reconsider throughput metrics and develop alternatives that better balance sequence length and accuracy, rewarding models that achieve comparable accuracy with meaningful generations.

It is also important to note that, to align with previous work, the benchmarks in this section focus on **multi-shot, short-sequence** generation. In these settings, the prompt constitutes the majority of the sequence, meaning suffix attention accounts for only a small fraction of the total computation. Per Amdahl's law (Amdahl, 1967), the maximum achievable speedup is therefore inherently bounded. Despite this limitation, *DPad* consistently delivers stable latency improvements even without system-level optimizations. Its true potential emerges in **low-shot, long-sequence** settings (discussed in Sec. 4.3.1), where the suffix becomes a more significant computational component, allowing *DPad* to achieve substantial additional latency reductions.

**Accuracy.** In addition to improving inference efficiency, *DPad* also enhances accuracy across nearly all tasks for the LLaDA models (Tbl. 2 and Tbl. 6), thereby defying the typical trade-off between speed and accuracy. For instance, *DPad* yields substantial gains in strict-match accuracy on GSM8K (+26.46%) and MATH (+19.62%) for LLaDA-Instruct. This improvement in strict-match score is particularly noteworthy, as it highlights *DPad*'s ability to enhance in-context learning. The vanilla backbone typically exhibits low strict-match performance (e.g., only 37.38% on GSM8K for LLaDA-Instruct), since this metric requires the model not only to produce the correct final answer (Flexible-Match) but also to adhere to the specific reasoning format demonstrated in few-shot exemplars, as illustrated in the Appx. E (Fig. 16). We posit that failures in strict matching often stem from interference by distant suffix tokens, which introduce low-value or off-format patterns that distract the model and encourage verbose or poorly structured outputs. By reducing the influence of such suffix tokens and directing attention toward high-value, information-rich prefix exemplars, *DPad* enables the model to replicate the structured reasoning formats required by strict matching more faithfully. In addition, by suppressing redundant generations, *DPad* facilitates earlier convergence to concise, well-formatted outputs, further improving strict-match performance without altering model parameters.

By contrast, the Dream model shows less advantage, with accuracy broadly comparable to the baseline and fluctuating only within a narrow margin. We attribute this discrepancy to the models' distinct training protocols. Since LLaDA is trained as a diffusion model from scratch, it learns to rely on the full suffix context during pre-training. Conversely, Dream is initialized from Qwen2.5 (Qwen et al., 2025), a pre-trained autoregressive model that has no exposure to suffix tokens. We therefore hypothesize that LLaDA's native dLLM architecture is inherently more sensitive to the suffix context, allowing it to benefit more significantly from the regularization that *DPad* provides.

## 4.3 Ablations and Analysis

### 4.3.1 Maximum Generation Length

To quantify efficiency gains in long-sequence generation, we analyze the speedups of various acceleration strategies on LLaDA-1.5/GSM8K (Tbl. 3). Our setup evaluates combinations of *DPad*, parallel decoding (`+Par.`), and prefix caching (`+PC.`) to disentangle the contribution of each technique. These findings are confirmed by a complementary analysis on the Dream-Base model (Appx. C.2.2).

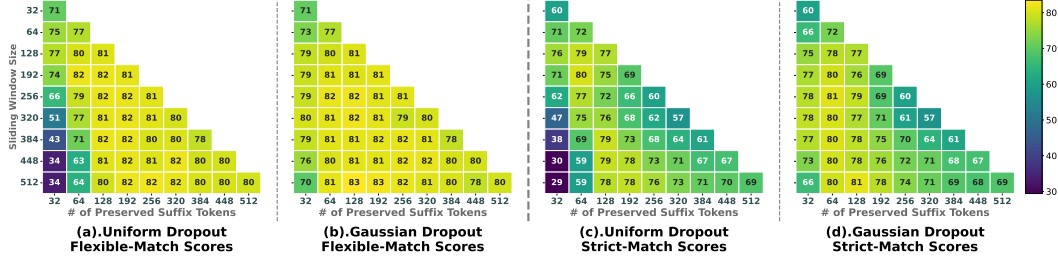

Figure 5: Ablation Study on Sliding Window Size and Dropout Function for *DPad* on LLaDA-1.5/GSM8K (512, 4-shot). Heatmaps showing Flexible-Match Accuracy scores with (a) uniform and (b) Gaussian dropout, and Strict-Match Accuracy scores with (c) uniform and (d) Gaussian dropout, across varying sliding window sizes and number of preserved suffix tokens. The (tokens, window_size) = (512, 512) configuration corresponds to the baseline, as it involves no token dropout.

**Efficiency.** We find that the acceleration benefits of *DPad* grow substantially with generation sequence length. As shown in Appx. C.1, for LLaDA-1.5 on GSM8K, improvements are modest at shorter sequence lengths (up to $1.51\times$ under a 256-token limit). However, when the maximum length is extended to 1024 tokens (single-shot setting), standalone *DPad* achieves a dramatic $\mathbf{20.3\times}$ speedup. Finally, when combined with parallel decoding and prefix caching (Fast-dLLM), the efficiency gains compound, yielding an overall $\mathbf{61.39\times}$ speedup compared to vanilla LLaDA and a $\mathbf{5.21\times}$ improvement over Fast-dLLM alone. These results demonstrate that suffix dropout and parallel decoding address orthogonal bottlenecks, and their combination enables significant improvements in long-sequence generation. A deeper analysis of these effects is provided in Appx. C.2.1.

Table 3: Performance on LLaDA-1.5 with GSM8K (1024 tokens, 1-shot).

| Method | Acc. (Flex. / Str.) | Lat.(s) / TPS | Speedup |
|---|---|---|---|
| Vanilla | 78.17 / 48.98 | 127 / 1.55 | 1.00× |
| + *DPad* | 78.77 / **74.07** | **6.28** / 18.4 | **20.3×** |
| +Par. | 78.77 / 49.43 | 11.7 / 16.9 | 1.00× |
| + *DPad* | 79.38 / **74.22** | **2.26** / 51.4 | **5.17×** |
| +Par. + PC. | 78.77 / 51.63 | 10.8 / 18.2 | 1.00× |
| + *DPad* | 77.10 / **70.66** | **2.07** / 55.5 | **5.21×** |
| Overall (+*Par.*+*PC.*+*DPad vs. Vanilla*) | | | **61.39×** |

**Accuracy.** Low-shot settings further highlight *DPad*'s ability to preserve and even enhance model accuracy. Compared with Tbl. 6, the strict-match score of the baseline drops significantly when the shots reduces from 4 to 1 (from ∼60% to ∼50%), whereas *DPad*'s performance remains remarkably stable (dropping only from ∼78% to ∼74%). This resilience demonstrates that *DPad* substantially strengthens the model's in-context learning capability, a significant achievement for a training-free method.

### 4.3.2 SLIDING WINDOW SIZE AND DROPOUT FUNCTION

We conducted an ablation study on LLaDA-1.5/GSM8K to determine the optimal sliding window size and dropout function for *DPad*, with results shown in Fig. 5.

Our analysis identifies a critical context window of 64-128 tokens immediately following the current block. The key principle for maximizing performance is to maintain a high density of preserved tokens within this critical window. Spreading a limited token budget (e.g., fewer than 128) thinly across a larger window was found to be counterproductive, significantly degrading accuracy.

The study also validates our choice of a **Gaussian dropout function over a uniform baseline**. While the two perform comparably with large token budgets, Gaussian dropout is consistently superior under low-budget conditions, especially when the sliding window is large. This is because it more efficiently allocates the limited token budget by prioritizing nearby tokens, a strategy that aligns with the consistent decaying patterns observed in Rotary Positional Embeddings (RoPE) (Su et al., 2021), attention scores (Fig. 4), and token confidence maps (Gong et al.).

Of course, Gaussian sampling may not be the optimal decay function, and other decay-based schemes (e.g., exponential, linear, or step-wise cutoff) remain to be explored. Nevertheless, in the training-free setting, we find that results are largely insensitive to the exact decay form, as long as the scheme emphasizes nearer tokens.

## 5 CONCLUSION

We addressed the high cost of full suffix attention in dLLMs—a key bottleneck caused by redundant computation—by introducing the Diffusion Scratchpad (*DPad*), a simple, training-free inference strategy. *DPad* leverages the inherent sparsity of suffix attention, combining a fixed-length sliding window with a distance-decay dropout that prunes low-value tokens *a priori*. This "winning ticket" approach yields up to a $61.4\times$ speedup when combined with existing optimizations, without sacrificing accuracy. Our results establish *DPad* as a practical and scalable solution that helps advance dLLMs from a promising alternative to a viable foundation for future language technologies.

### ETHICS STATEMENT

This research focuses on improving the efficiency of diffusion-based large language models (dLLMs) through a training-free inference strategy (*DPad*). Our study does not involve human subjects, user data, or personally identifiable information. All datasets used in evaluation (GSM8K, MATH, HumanEval, MBPP) are publicly available and widely adopted benchmarks, cited appropriately in the paper.

**Bias and fairness.** As our method is applied to pre-existing models and datasets, any biases or limitations originate from the underlying training corpora of the base models and benchmarks. We do not introduce new datasets, but we acknowledge that accelerated inference could facilitate broader deployment of LLMs, which inherit and may propagate these biases.

**Privacy and safety.** Our work does not involve private data. The proposed method is model-agnostic and does not alter the underlying training distribution.

**Dual use considerations.** Improvements in inference efficiency may enable faster and more widespread use of LLMs. We encourage responsible application, with attention to fairness, transparency, and safety in downstream use.

### REPRODUCIBILITY

We provide all resources needed to reproduce our results. Implementation details, hyperparameters, and ablation setups are described in Section 3 and 4 and Appendices B and C. Specifically: (i) code and scripts for suffix dropout (*DPad*) and evaluation will be released with instructions for running on each benchmark; (ii) exact hyperparameter settings (sliding window size, decay rate, scaling factor) are reported in Section 4.1 and Appendix B; (iii) dataset usage (GSM8K, MATH, HumanEval, MBPP) follows standard publicly available versions with citations; (iv) experiments were run on NVIDIA A100 80GB GPUs, as noted in Section 4.1; (v) random seeds and evaluation commands are documented in the code release. An environment will also be provided as "requirements.txt" . All tables and figures can be regenerated directly from the released code and instructions, available as supplementary material.

### ACKNOWLEDGMENTS

This work was supported in part by NSF-2112562, NSF-2328805, and ARO W911NF-23-2-0224. The authors sincerely thank the anonymous reviewers for their constructive feedback and valuable suggestions that greatly improved the quality of this work.

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

# A OBSERVATIONS

## A.1 EMPIRICAL VERIFICATION OF THE SCRATCHPAD MECHANISM

In Section 3.1, we hypothesized that suffix tokens function as an *Attention Connection*—a "scratch-pad" that stores intermediate states to guide denoising. To verify this, we analyze the scratchpad mechanism along two dimensions: (1) **Cross-Layer** (transmitting information across Transformer layers within a step) and (2) **Cross-Step** (transmitting information between decoding steps).

We quantify the criticality of these dimensions by evaluating performance on *Countdown* (Ye et al., 2024) and *Sudoku* ($4 \times 4$) (Seely et al., 2025)—tasks requiring strict global consistency and thus serving as ideal stress tests for the scratchpad mechanism—across a spectrum of scratchpad availability:

- **No Scratchpad (ARMs):** We evaluated LLaMA-3.1-8B-Instruct (Grattafiori et al., 2024) and Qwen-2.5-7B (Qwen et al., 2025). As ARMs inherently lack the bidirectional attention required to attend to future tokens, they are structurally blind to the suffix and serve as a baseline for the complete absence of a scratchpad.

- **Partial Scratchpad (Semi-Autoregressive dLLMs):** We evaluated standard Semi-AR decoding and **Semi-AR with Suffix Dropout (Semi-AR+SD)**. In this setting, suffix tokens are masked between steps, prohibiting *Cross-Step* information transmission. However, the model retains a limited scratchpad capability via *Cross-Layer* attention within the current block.

- **Full Scratchpad (Global-Diffusion dLLMs):** Under Global-Diffusion decoding, the model writes to confident suffix tokens, enabling both *Cross-Layer* and *Cross-Step* scratch-pad functions. Alongside the Global-Diffusion baseline, we introduce **Semi-AR+Suffix Dropout+Pre-Decoding (Semi-AR+SD+PD)**. In this variant, preserved suffix tokens are temporarily unmasked during the current block's decoding (conditioned on confidence) and remasked upon completion of current block. This effectively restores the *Cross-Step* transmission channel.

Table 4: Representative examples of the Countdown (Ye et al., 2024) and Sudoku (Seely et al., 2025) benchmarks. Sudoku ($4 \times 4$) serves as a rigorous test for global planning, requiring the model to fill empty cells ('0') while maintaining strict consistency across rows, columns, and subgrids.

| Task | Question | Answer |
|------|----------|--------|
| **Countdown** | Given 5 numbers, use +-*/ to operate over the first four numbers to achieve the fifth number.
Input: 81,84,82,42,41
Output: | 82-81=1, 84-1=83, 83-42=41 |
| **Sudoku** | Fill the position where the values are 0 in a 4x4 grid with digits 1-4 so that each column, each row, and each of the four 2x2 subgrids that compose the grid contains all of the digits from 1 to 4.
Input:
4001
3124
1302
0400
Output: | 4231
3124
1342
2413 |

The results (summarized in Tbl. 5) reveal a decisive behavioral divergence. On Sudoku, both the No Scratchpad (ARMs) and Partial Scratchpad (Semi-AR / Semi-AR+SD) groups achieve $0.0$ accuracy,

Table 5: Quantitative verification of the scratchpad mechanism on planning benchmarks. We compare performance across three levels of memory availability. The results show that Sudoku performance collapses to 0.0 without the full scratchpad (AR models and Semi-AR dLLMs) and recovers only when the suffix memory is fully enabled (Semi-AR+SD+PD and Global-Diffusion).

| Scratchpad Setting | Models | Countdown | Sudoku |
|---|---|---|---|
| No Scratchpad | LLaMA-3.1-8B-Instruct | 27.0 | 0.0 |
| | Qwen-2.5-7B | 0 | 0 |
| Partial Scratchpad (Only *cross-layer*) | LLaDA-Instruct (Semi-AR) | 32.6 | 0 |
| | LLaDA-Instruct (Semi-AR+SD) | 32.6 | 0 |
| Full Scratchpad (*cross-layer + cross-step*) | LLaDA-Instruct (Global-Diffusion) | 34.7 | 24.6 |
| | LLaDA-Instruct (Semi-AR+SD+PD) | 32.6 | 24.4 |

indicating a complete failure to handle global consistency without cross-step memory. In contrast, the Full Scratchpad group demonstrates superior performance: Global-Diffusion LLaDA achieves scores of 34.7% on Countdown and 24.6% on Sudoku. Crucially, restoring the cross-step scratchpad via *Semi-AR+SD+PD* boosts Sudoku performance from 0.0 to 24.4%, effectively matching the Global-Diffusion baseline (24.6%). This confirms that the *Cross-Step* scratchpad is the critical component for global planning.

## A.2 ANALYSIS OF SUFFIX TOKEN ENTROPY

To provide further intuition for our method, we analyze the *Local Entropy* of generated suffix tokens. It is important to distinguish this metric from standard predictive entropy (uncertainty over the vocabulary). Here, we define Local Entropy as the statistical diversity of tokens generated at a specific suffix position across multiple samples. High local entropy indicates a position that changes dynamically depending on context (e.g., a reasoning step), while low local entropy indicates a position that remains static.

As shown in Figure 6, the entropy of suffix tokens decays rapidly with distance from the current generation block. Tokens far into the suffix exhibit near-zero entropy, meaning they do not vary across samples. Qualitative analysis confirms that these are not deterministic structural elements (which would shift based on content length), but rather static, repetitive patterns (e.g., repeated `<eos>` tokens or padding). This empirical finding strongly motivates our distance-decay dropout strategy, which prioritizes pruning these low-value, redundant tokens.

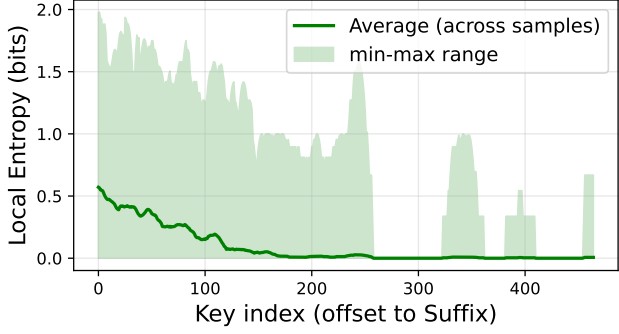

Figure 6: **Local entropy of suffix tokens.** The plot shows the average local entropy (in bits) of suffix tokens as a function of their distance from the current block. Entropy rapidly decays, approaching zero for distant tokens. This confirms that distant suffix tokens are highly redundant and can be pruned with minimal information loss, validating the core assumption of *DPad*.

## Prompt

Janet's ducks lay 16 eggs per day. She eats three for breakfast every morning andbakes muffins for her friends every day with four. She sells the remainder at thefarmers' market daily for $2 per fresh duck egg. How much in dollars does she makeevery day at the farmers' market?

### Block 0, Step 7

Ċ To determine how much Janet makes every day at the farmers ' market , we need to follow these steps : Ċ Ċ 1 . Calculate Calculate the number of eggs eggs per day day day Ċ Ċ Ċ Ċ Ċ  6 6 per per per Ċ Ċ Ċ Ċ Ċ Ċ Ċ Ċ Ċ Ċ Ċ Ċ Ċ Ċ Ċ Ċ Ċ Ċ Ċ Ċ Ċ Ċ Ċ Ċ Ċ Ċ Ċ Ċ Ċ Ċ Ċ Ċ Ċ Ċ Ċ Ċ Ċ Ċ Ċ Ċ Ċ Ċ Ċ Ċ Ċ Ċ Ċ Ċ Ċ Ċ Ċ Ċ Ċ Ċ Ċ Ċ Ċ Ċ Ċ Ċ Ċ Ċ Ċ Ċ Ċ Ċ Ċ Ċ Ċ Ċ Ċ Ċ Ċ Ċ Ċ Ċ  Ċ Ċ   Ċ Ċ Ċ Ċ Ċ Ċ Ċ Ċ Ċ Ċ Ċ Ċ Ċ Ċ Ċ Ċ Ċ Ċ Ċ Ċ Ċ Ċ Ċ Ċ Ċ Ċ Ċ Ċ Ċ Ċ Ċ Ċ Ċ Ċ Ċ Ċ Ċ Ċ Ċ Ċ Ċ Ċ Ċ Ċ \\\\\\\\\\\\\\\\{ \ \ =  \ \ \ dollars dollars dollars Ċ Ċ Ċ Ċ Ċ Ċ Ċ makes makes makes makes 1 day day day at the market ' market . Ċ Ċ  1 4 Ċ <| endoftext|>

### Final Answer

Ċ To determine how much Janet makes every day at the farmers ' market , we need to follow these steps : Ċ Ċ 1 . Calculate the total number of eggs laid by Janet 's ducks per day Ċ Ċ 2 . Sub tract the number of eggs Janet eats for breakfast and b akes for her friends . Ċ 3 . Multiply the remaining eggs by the price per egg to find her daily earnings Ċ Ċ Ċ 1 . Janet 's ducks lay 16 eggs per day . Ċ 2 . Janet eats 3 eggs for breakfast and b akes 4 eggs for her friends . Therefore , she uses a total of 3 + 4 = 7 eggs per day Ċ Ċ 3 . The number of eggs left for sale is 16 - 7 = 9 eggs . Ċ 4 . Janet sells each egg for $ 2 ' Therefore , her daily earnings from selling the eggs are 9 * $ 2 = $ 18 Ċ Ċ Ċ So , Janet makes $ 18 every day at the farmers ' market Ċ Ċ #### 18 <|eot_id|> <|endoftext|>

Current Tokens   Suffix Tokens   Ċ: Newline

Figure 7: **Decoding of response tokens at an intermediate step**. The plot shows the most likely tokens predicted by the model at a given decoding stage, reflecting its intermediate understanding and behavior in response to the prompt. Notably, the model repeatedly records key tokens in the suffix, irrespective of position, revealing a scratchpad-like pattern that facilitates decoding of the current block.

### A.3 PRE-DECODED SUFFIX TOKENS

To further illustrate the scratchpad mechanism, we present the most likely tokens that LLaDA 1.5 (Zhu et al., 2025) predicts at both the current and suffix positions, as shown in Figure 7 (bottom left). The sample question is drawn from GSM8k (Cobbe et al., 2021), and a final generated response is included for reference. From the intermediate predictions, we observe a recurring pattern where key tokens are repeatedly placed across multiple suffix positions. This behavior resembles the use of a scratchpad, where the model leverages available "empty" positions to temporarily store useful information that aids in decoding the current tokens.

### A.4 SUFFIX TOKENS ATTENTION SCORE

We visualize the attention scores of layers 0, 7, 15, 23, and 31 in LLaDA-1.5, following the same procedure described in Fig. 3. The results are shown in Fig. 8.

Across layers, we observe a consistent decay in attention paid to suffix tokens as their positions move farther from the current block. In addition, attention spikes on suffix tokens become increasingly pronounced in deeper layers, suggesting that the model allocates more focused attention to these suffix tokens as the forward pass progresses.

### A.5 GENERALIZATION OF SCRATCHPAD MECHANISM TO DREAM MODELS

To verify the generalization of the scratchpad mechanism to other dLLM architectures, we extended our analysis (specifically, the Attention Score Map and Spike Shift experiments) to the Dream-Base model.

Fig. 9 illustrates the attention score distribution for Dream. Similar to LLaDA, Dream exhibits a consistent distance-decay trend. However, compared to LLaDA, Dream displays significantly fewer high-attention spikes in the suffix region. In many layers, the attention scores allocated to suffix tokens are near-zero. We hypothesize that this behavior stems from Dream's training paradigm: it

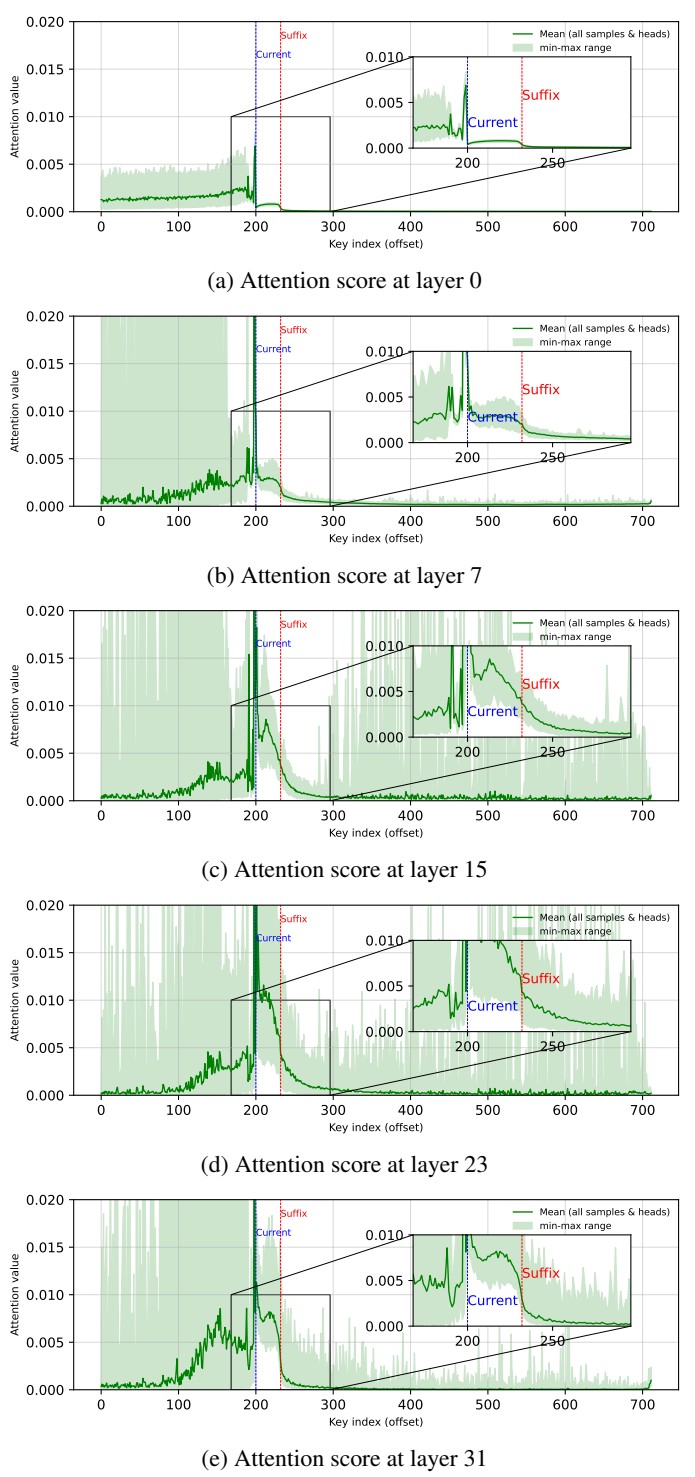

(a) Attention score at layer 0

(b) Attention score at layer 7

(c) Attention score at layer 15

(d) Attention score at layer 23

(e) Attention score at layer 31

Figure 8: Attention score at different layers in LLaDA-1.5 on GSM8K with 512 generation length.

is initialized from Qwen-2.5, a standard autoregressive model, which inherently lacks bidirectional attention to future tokens during pre-training.

Despite this general sparsity, we observed distinct attention spikes in deeper layers (e.g., Layer 23). To test the robustness of these features, we conducted the same spike shift experiment on the Dream-Base model. Notably, the "spike shift" phenomenon is replicated in Dream, as shown in Fig. 10.

When a high-attention suffix token is forcibly pruned, the model shifts its attention to an adjacent neighbor (e.g., shifting from index 50 to 54), thereby preserving the informational flow.

The replication of both distance-decay patterns and position-adaptive attention shifts strongly supports the generalization of the Scratchpad mechanism and the Diffusion Lottery Ticket hypothesis across different dLLM backbones.

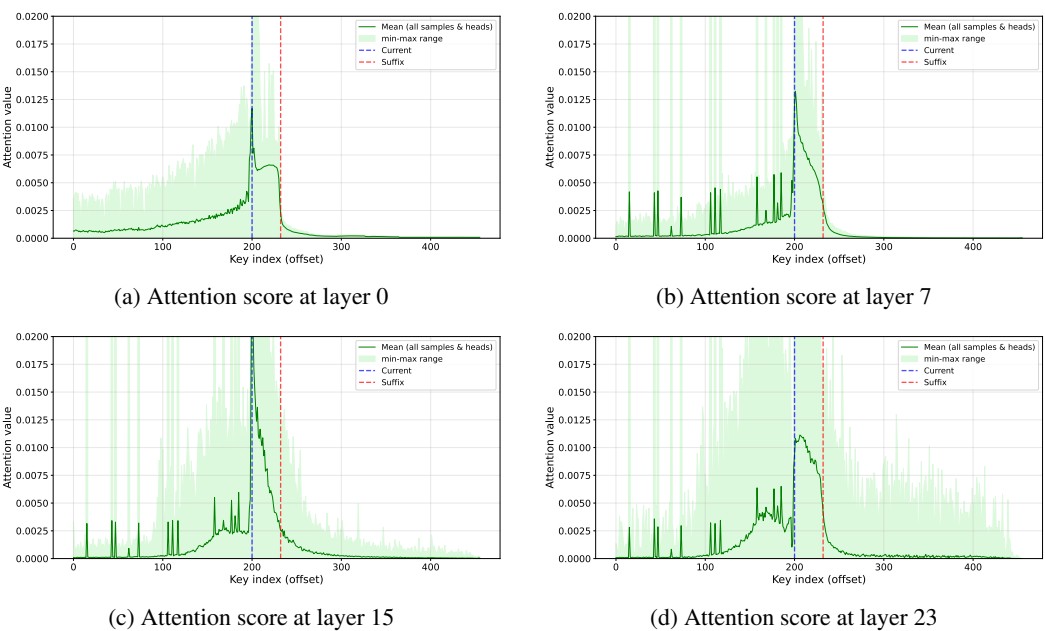

(a) Attention score at layer 0      (b) Attention score at layer 7

(c) Attention score at layer 15      (d) Attention score at layer 23

Figure 9: Attention score at different layers in Dream-Base on GSM8K with 256 generation length.

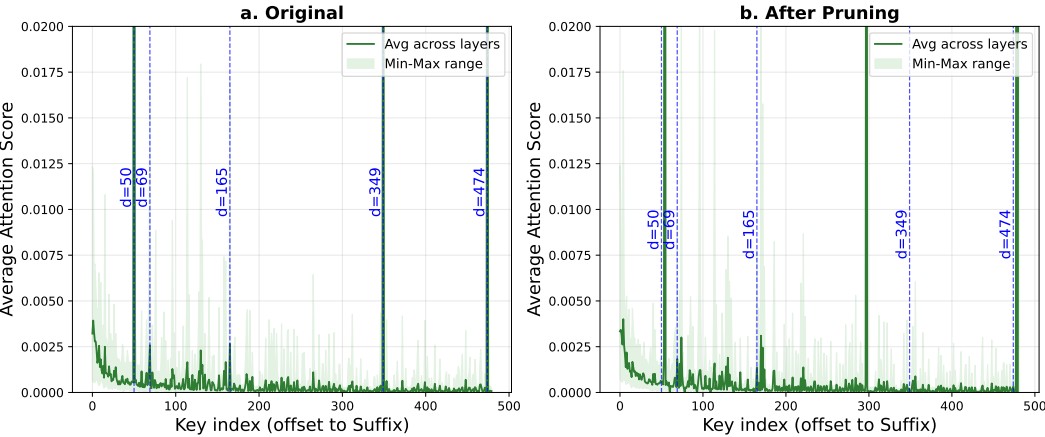

Figure 10: (a) Attention scores of suffix tokens paid by current block tokens ($A[c{:}s,\ s{:}]$) across layers in Dream-Base, showing overall decay with occasional spikes (e.g., d = 50, 349, 474). (b) After forcibly pruning these spike positions, attention shifts to nearby tokens, indicating that adjacent positions can absorb suffix information (e.g., pruning token 50 shifts the spike to token 54).

# B  IMPLEMENTATION DETAILS

This section provides implementation details for *DPad*, including the core code logic, the early termination mechanism, and the necessary adjustment to RoPE to ensure positional consistency.

This dropout strategy is not static but follows a block-wise schedule (Nie et al.). As shown in Fig. 1 (c), after each block is generated, all previously dropped tokens are restored, and Gaussian sampling is re-applied to the updated suffix set before decoding the next block. This dynamic re-sampling prevents sampling bias and ensures that no token is systematically ignored.

**Core Logic.** *DPad*'s core mechanism can be implemented with only a few lines of Python code. The primary operation involves sampling a subset of suffix token indices using a Gaussian distance-decay function and then forwarding only this pruned sequence to the model.

---

**Algorithm 1** *DPad* Suffix-Dropout Inference Step

**Input:** Indices of tokens $\in x : \mathcal{P}$ (prefix), $\mathcal{C}$ (current block), $\mathcal{S}$ (suffix); window $W$; decay params $(k, a)$; model $p_\theta$
**Output:** model output for this step
 1: */*select a near-suffix window */
 2: $\mathcal{S}_W \leftarrow$ the first $W$ indices of $\mathcal{S}$
 3: /* Select suffix indices using distance-decay sampling */
 4: $\mathcal{K} \leftarrow \{j \in \mathcal{S}_W \mid u_j \le P(d_j)\}$
         where $d_j = j - \max(\mathcal{C}) - 1$ (dist to current), $u_j \sim \mathrm{Uniform}(0, 1)$,
         and $P(d_j) = \min\{1, a(2\pi)^{-1/2} \exp(-\frac{1}{2}(\frac{k}{W} d_j)^2)\}$
 5: $\mathsf{scratchpad} \leftarrow x[\mathcal{K}]$
 6: $x' \leftarrow \mathrm{concat}(x[\mathcal{P}], x[\mathcal{C}], \mathsf{scratchpad})$
 7: $\mathsf{output} \leftarrow p_\theta(x')$
 8: **return** output

---

As outlined in Algorithm 1, the Gaussian-based sampler selects a subset of suffix tokens according to distance-aware probabilities. We then construct a pruned sequence $x'$ by indexing with $\mathcal{K}$. The rest of the model remains unchanged, except for a minor adjustment to the RoPE embeddings inside the attention module to ensure correct positional encoding under suffix dropout.

**RoPE Adjustment.** Our suffix dropout mechanism requires only a minor adjustment to the RoPE (Su et al., 2021) to maintain correct positional information. In standard RoPE, a token at absolute position $i$ is encoded using an angle $\theta_i = i \cdot \Delta$. After dropout, however, only a sparse, non-contiguous subset of suffix indices $\mathcal{I} = \{i_1, \dots, i_m\}$ is preserved.

To handle this, we ensure that each preserved token retains its original positional information. Rather than using their re-indexed positions after dropout, we apply a mapping function $f(i_k)$ that retrieves the original absolute position of the $k$-th preserved token. The new angle is then computed as

$$\theta'_{i_k} = f(i_k) \cdot \Delta. \tag{8}$$

Accordingly, the modified RoPE application becomes

$$\mathrm{RoPE}'(\mathbf{x}_{i_k}, i_k) = \mathrm{RoPE}(\mathbf{x}_{i_k}, f(i_k)). \tag{9}$$

This adjustment requires only a lightweight remapping inside the attention module and does not alter the functional form of RoPE, confirming that suffix dropout is nearly cost-free while preserving positional consistency.

**Early Termination.** To further enhance efficiency, we also implement an simple technique, called early termination. It halts generation upon detecting an end-of-sequence (<eos>) token. This technique, which performs a check after each decoded block, addresses the computational redundancy inherent in fixed-length generation models like LLaDA (Nie et al.). Its utility is most pronounced in long-sequence settings where the maximum length significantly exceeds the actual generated content, particularly for models employing top-k decoding.

Conversely, the benefits of early termination are marginal when paired with parallel decoding, as that method already minimizes post-`<eos>` redundancy by unmasking each remaining block in a single step. Similarly, the technique is less impactful for Dream models, which tend to fully utilize their token budget, as evidenced by $\bar{\ell}/\ell_{\max}$ in Tbl. 2. Therefore, to establish a more robust point of comparison, we applied this optimization exclusively to the LLaDA `vanilla` baseline in our long-sequence analysis, as discussed in Appx. C.2.1.

## C EXTENDED EXPERIMENTAL RESULTS

This section provides supplementary experimental results, including a detailed performance breakdown for the LLaDA-1.5 model and an extended analysis of long-sequence generation on the Dream-Base model.

### C.1 FULL PERFORMANCE RESULTS FOR LLaDA-1.5

We present a comprehensive analysis of *DPad*'s performance on the LLaDA-1.5 model across all benchmarks. The full results are summarized in Tbl. 6.

Table 6: Performance of LLaDA-1.5 with *DPad* on four benchmarks.

| Benchmark | Method | Efficiency | | | | | Accuracy (%) | |
|---|---|---|---|---|---|---|---|---|
| | | Latency(s)↓ | | TPS↑ | | $\bar{\ell}/\ell_{\max}$ | Flexible↑ | Strict↑ |
| GSM8K *4-shot* | Vanilla | 27.61 | 1.00× | 7.77 | 1.00× | 215 / 256 | 80.59 | 61.87 |
| | **+*DPad*** | 18.26 | 1.51× | 8.56 | 1.10× | 156 / 256 | 80.14 | 78.47 |
| | +Parallel (Fast-dLLM) | 8.06 | 3.42× | 26.61 | 3.43× | 215 / 256 | 80.82 | 62.62 |
| | +Parallel+*DPad* | **6.23** | **4.43×** | 25.23 | 3.25× | 157 / 256 | 80.89 | 78.92 |
| MATH *4-shot* | Vanilla | 25.12 | 1.00× | 8.67 | 1.00× | 218 / 256 | 33.52 | 32.72 |
| | **+*DPad*** | 20.63 | 1.22× | 9.48 | 1.09× | 196 / 256 | 34.08 | 37.00 |
| | +Parallel (Fast-dLLM) | 9.48 | 2.65× | 22.96 | 2.65× | 218 / 256 | 33.60 | 32.92 |
| | +Parallel+*DPad* | **8.57** | **2.93×** | 22.76 | 2.63× | 195 / 512 | 32.92 | 35.96 |
| HumanEval *0-shot* | Vanilla | 34.80 | 1.00× | 3.16 | 1.00× | 110 / 512 | 40.85 | – |
| | **+*DPad*** | 11.55 | 3.01× | 7.19 | 2.28× | 83 / 512 | 44.51 | – |
| | +Parallel (Fast-dLLM) | 11.16 | 3.12× | 9.80 | 3.10× | 109 / 512 | 39.63 | – |
| | +Parallel+*DPad* | **5.26** | **6.61×** | 15.64 | 4.95× | 82 / 512 | 39.63 | – |
| MBPP *3-shot* | Vanilla | 62.34 | 1.00× | 1.02 | 1.00× | 63 / 512 | 38.20 | – |
| | **+*DPad*** | 14.95 | 4.17× | 4.33 | 4.26× | 65 / 512 | 39.80 | – |
| | +Parallel (Fast-dLLM) | 5.47 | 11.39× | 11.62 | 11.44× | 64 / 512 | 38.60 | – |
| | +Parallel+*DPad* | **4.41** | **14.14×** | 14.83 | 14.60× | 65 / 512 | 41.60 | – |

**Efficiency.** On LLaDA-1.5, *DPad* achieves significant latency reductions, with speedups ranging from **1.22×** to **4.17×** over the vanilla baseline. When combined with parallel decoding, the overall speedup reaches up to **14.14×**. These gains are driven by *DPad*'s core mechanisms, including the generation of more concise outputs. For instance, *DPad* reduced the average generation length on MATH by approximately 10% and on GSM8K by 27%, with the majority of the speedup still attributable to reduced suffix computation. As shown in Figure 11, the benefits of *DPad* scale dramatically with sequence length, achieving a **61.39×** combined speedup in a 1024-token setting.

**Accuracy.** Consistent with results on LLaDA-Instruct, *DPad* improves both efficiency and accuracy for LLaDA-1.5, demonstrating that speed enhancements do not compromise quality. The most substantial improvements are observed in strict-match accuracy, where *DPad* helps the model better adhere to the formatting of in-context examples. On GSM8K, *DPad* improves the strict-match score from 61.87% to 78.47%. This highlights *DPad*'s ability to focus the model on high-quality signals from the prompt by pruning noisy, low-entropy suffix tokens.

## C.2 FULL LONG-SEQUENCE GENERATION ANALYSIS

### C.2.1 LLADA-1.5

The $20.3\times$ speedup of *DPad* over the vanilla baseline reported in Tbl. 3 warrants further analysis, as it is disproportionately large compared to the $5.2\times$ speedups over the +Par. and +Par.+PC. baselines. Upon investigation, we find this discrepancy is primarily due to the vanilla model's inefficiency: it often produces a concise answer (around 256 tokens) but then continues to generate redundant <eos> tokens one by one to fill the 1024-token budget, a process that incurs significant latency. In contrast, our *DPad* implementation eliminates this specific inefficiency by incorporating the early termination technique discussed in Appx. B.

To disentangle the individual contributions of suffix dropout and early termination, we introduce a stronger baseline by augmenting the vanilla model with an early termination mechanism (+Early Termination). As shown in Fig. 11, our analysis reveals that a $4.8\times$ speedup is attributable to early termination alone. Nevertheless, even against this stronger baseline, suffix dropout contributes an additional $4.2\times$ speedup, demonstrating its significant and independent impact on efficiency.

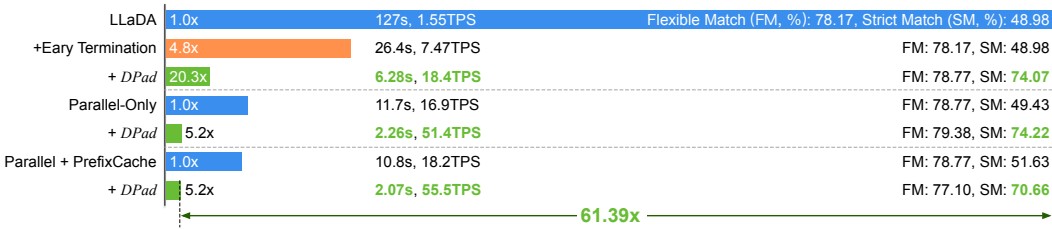

Figure 11: Latency comparison on LLaDA-1.5 with GSM8K (1024 tokens, 1-shot).

### C.2.2 DREAM

To further validate the scalability of our approach, we conducted an extended analysis on the Dream-Base model using the HumanEval benchmark. We evaluated the same set of acceleration strategies as in the main paper's analysis. The results, shown in Figure 12, confirm that the benefits of *DPad* generalize across different models and tasks.

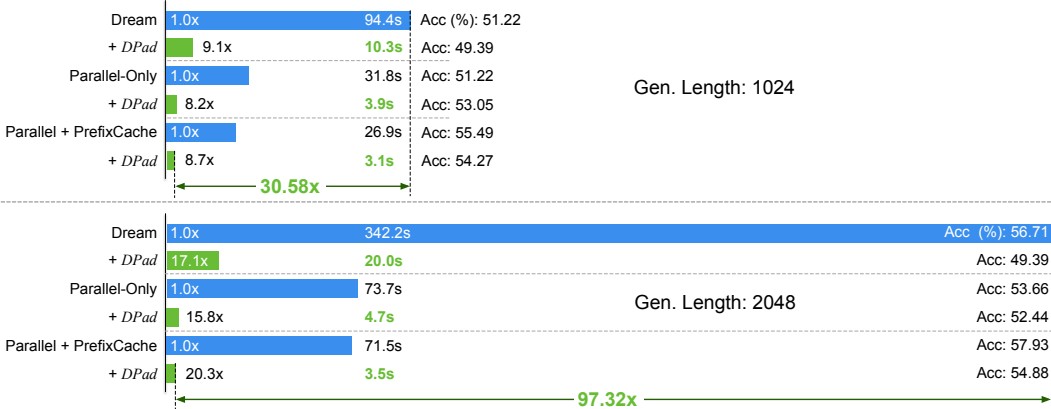

Figure 12: ***DPad* performance on long-sequence generation with Dream-Base.** Latency and accuracy comparison on the HumanEval benchmark for generation lengths of 1024 and 2048 tokens (0-shot). *DPad*'s speedup scales significantly with sequence length, achieving up to a $97.32\times$ speedup when combined with other optimizations at 2048 tokens.

**Speedup.** We observe a similar scaling trend on Dream-Base as on LLaDA-1.5. *DPad* alone accelerates inference by $9.1\times$ for 1024-token sequences and $17.1\times$ for 2048-token sequences. When

combined with Fast-dLLM, the benefits are multiplicative, achieving a **30.58×** speedup at 1024 tokens and an impressive **97.32×** speedup at 2048 tokens. These results reinforce that suffix dropout and parallel decoding address orthogonal bottlenecks, yielding nearly two orders of magnitude improvement in long-sequence generation.

**Accuracy.** On the 2048-token HumanEval task, we observed a 7.32% accuracy degradation when applying *DPad* to the vanilla Dream model. This degradation is largely mitigated when *DPad* is combined with Fast-dLLM. We hypothesize that this isolated performance drop arises from complex interactions between our training-free pruning strategy and the model's native Top-$k$ sampling behavior, pointing to an interesting direction for future investigation.

## C.3 COMPARISON WITH SPARSE-DLLM

Recent works such as Sparse-dLLM (Song et al., 2025a) introduce a token-pruning strategy for dLLMs inspired by the "compute-and-evict" mechanisms of autoregressive models. While both approaches aim to prune unimportant tokens, the core difference lies in *when* and *how* pruning occurs: Sparse-dLLM employs a *reactive* eviction method, whereas *DPad* utilizes a *proactive* dropout method.

### C.3.1 OPERATIONAL PARADIGM AND MEMORY COMPLEXITY

**Proactive vs. Reactive Pruning.** Sparse-dLLM essentially transplants the conventional "compute-and-evict" strategy from autoregressive models (Song et al., 2025a) directly to the dLLM context. As a reactive method, it must first compute the full attention matrix to identify salient tokens before evicting the unimportant ones. This generic approach incurs computational overhead and, crucially, requires **high peak memory allocation** to process the full sequence prior to eviction.

In contrast, *DPad* diverges from this adaptation by exploiting a unique property intrinsic to dLLMs: the DLT Hypothesis (Sec. 3.3). By using distance as a reliable proxy for utility *before* attention is computed, *DPad* renders pruned tokens "invisible" to the computational graph. This ensures **zero runtime overhead** for the pruning step itself, demonstrating a design tailored specifically for diffusion generation rather than transplanted from autoregressive paradigms.

**Memory Scaling.** Regarding memory complexity, since the Prefix-Sparse strategy (Song et al., 2025a) is compatible with both methods and does not differentiate them, we focus on the handling of suffix tokens. Sparse-dLLM typically requires a retention ratio of $\sim 50\%$(Song et al., 2025a) that scales linearly with the sequence length $L$, resulting in $O(L)$ complexity. Conversely, *DPad* employs a fixed sliding window with a more aggressive retention ratio (20%–37.5%, see Appx. D). Consequently, *DPad*'s memory footprint for the suffix is strictly bounded by the window size $W$, effectively achieving $O(1)$ complexity relative to the total sequence length. A summary of these differences is provided in Tbl. 7.

Table 7: Comparison of Operational Paradigm and Memory Complexity for Suffix Tokens.

| Method | Pruning Paradigm | Memory Complexity (Suffix) | Sparsity |
|---|---|---|---|
| Fast-dLLM | N/A (Dense) | $O(L)$ | 100% |
| Sparse-dLLM | Reactive (Post-Attention) | $O(L)$ | $\sim 50\%$ |
| *DPad* (Ours) | **Proactive (Pre-Attention)** | $O(1)$ | $\leq 20\% - 37.5\%$ |

### C.3.2 EMPIRICAL EVALUATION

To quantify the practical benefits of the proactive paradigm, we conducted a head-to-head comparison on 100 samples from the GSM8K benchmark using the LLaDA-1.5 model. We evaluated

performance on long-context scenarios ($l_{\max} \in \{2048, 4096\}$). To ensure a fair comparison, both the Sparse-dLLM baseline and our *DPad* method are implemented on top of the PrefixCache (PC) optimization (Wu et al.).

**Efficiency and Accuracy.** As shown in Tbl. 8, *DPad* demonstrates superior scaling for long sequences:

- **Latency:** *DPad* achieves a **12.87×** speedup over Sparse-dLLM at 2048 tokens. This advantage expands to **39.64×** at 4096 tokens, highlighting the benefits of $O(1)$ complexity scaling.

- **Throughput (TPS):** Similar to the observations in Sec. 4, the speedup in terms of TPS is more moderate compared to end-to-end latency. However, *DPad* still surpasses Sparse-dLLM. While Sparse-dLLM experiences throughput degradation at longer context lengths ($l_{\max} = 4096$) due to the linear complexity of its pre-eviction calculation, *DPad* maintains consistent throughput, achieving a **1.30×** speedup. This confirms that proactive pruning effectively alleviates the computational bottlenecks that constrain generation speed.

- **Accuracy:** In both settings, *DPad* maintains comparable or superior accuracy (e.g., **84.00%** vs. 79.00% at 2048 tokens), demonstrating that proactive pruning does not compromise generation quality.

Table 8: Efficiency and accuracy evaluation of Sparse-dLLM vs. *DPad* using LLaDA-1.5 on 100 samples of GSM8K with different max generation length ($l_{\max}$)

| | | LLaDA-1.5 | | | | |
|---|---|---|---|---|---|---|
| | | Efficiency | | | | Accuracy (%) |
| Context Length | Method | Latency(s)↓ | Speedup | TPS↑ | Speedup | Flexible↑ |
| $l_{\max} = 2048$ | Vanilla+PC+Sparse-dLLM | 83.30 | 1.00× | 24.28 | 1.00× | 79.00 |
| | Vanilla+PC+*DPad* | **6.47** | **12.87×** | **24.42** | **1.01×** | **84.00** |
| $l_{\max} = 4096$ | Vanilla+PC+Sparse-dLLM | 213.65 | 1.00× | 19.17 | 1.00× | **80.00** |
| | Vanilla+PC+*DPad* | **5.39** | **39.64×** | **24.96** | **1.30×** | 79.00 |

**Memory Stability.** Fig. 13 illustrates the memory footprint throughout the generation process. Attributable to its "cache-then-evict" policy, Sparse-dLLM exhibits a highly unstable active memory profile, characterized by frequent and pronounced oscillations (Fig. 13a).

Crucially, this reactive pruning renders the apparent active memory savings "illusory" from a system allocation perspective. As shown in Fig. 13b, Sparse-dLLM demands the same **Peak Memory** allocation as the dense Fast-dLLM baseline, because the system must reserve sufficient buffer capacity to compute the full attention matrix before eviction occurs. Consequently, if the system were to allocate the ostensibly "saved" memory to concurrent workloads, the primary process would encounter Out-Of-Memory (OOM) errors during the pre-eviction computation spikes.

In contrast, *DPad* operates proactively: it eliminates the need to cache pruned suffix tokens entirely. This results in strictly lower peak memory usage, yielding an approximate 50% reduction in memory resource requirements compared to the baselines. As a result, the memory footprint of *DPad* aligns closely with the stable growth pattern of standard autoregressive models, thereby facilitating efficient system-level memory allocation.

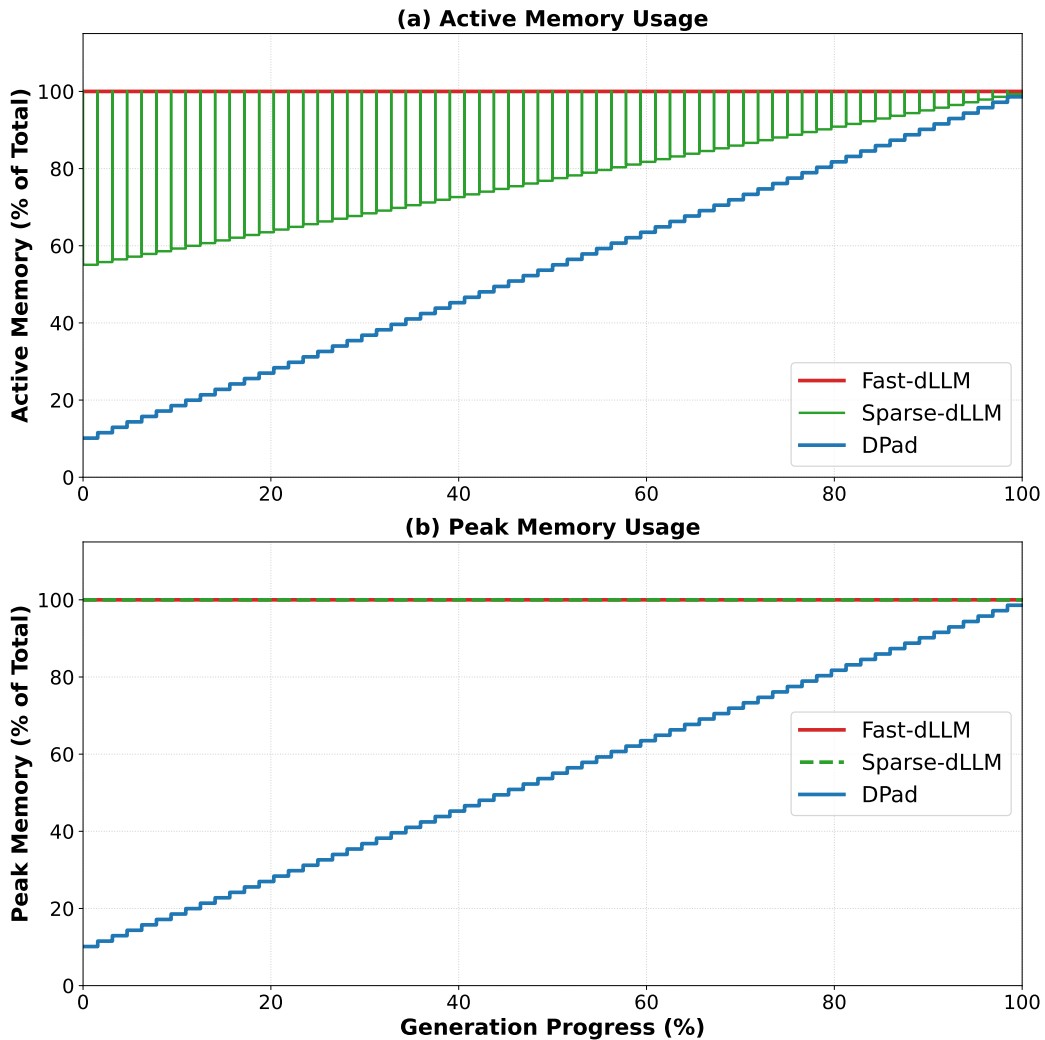

Figure 13: Memory Usage comparison on long-sequence generation. Fast-dLLM serves as the baseline (100%). To specifically isolate the impact of suffix token handling, neither Sparse-dLLM nor *DPad* utilizes the Prefix-Sparse technique in this comparison. (a) **Active Memory**: Sparse-dLLM shows unstable oscillations due to reactive eviction. (b) **Peak Memory**: Sparse-dLLM fails to reduce peak requirements compared to Fast-dLLM, whereas *DPad* achieves significant real-world memory savings.

## D    HYPERPARAMETER ANALYSIS

This section details the hyperparameter tuning process for our Gaussian sampler. Since attention score distributions can vary across models and datasets, we tune hyperparameters on a small subset of data for each benchmark to ensure optimal performance.

### D.1    ABLATION STUDY ON GAUSSIAN HYPERPARAMETERS

Inspired by findings that dLLMs exhibit different behaviors on mathematical and code-generation tasks (Gong et al.), we tune the hyperparameters for our Gaussian Sampler separately for each domain. We perform a grid search over two key parameters: the decay rate, $k$, and a scale factor, $a$. The parameter $a$ is used to control the overall retention density, which we define as the expected proportion of suffix tokens preserved by the Gaussian Sampler inside the sliding window.

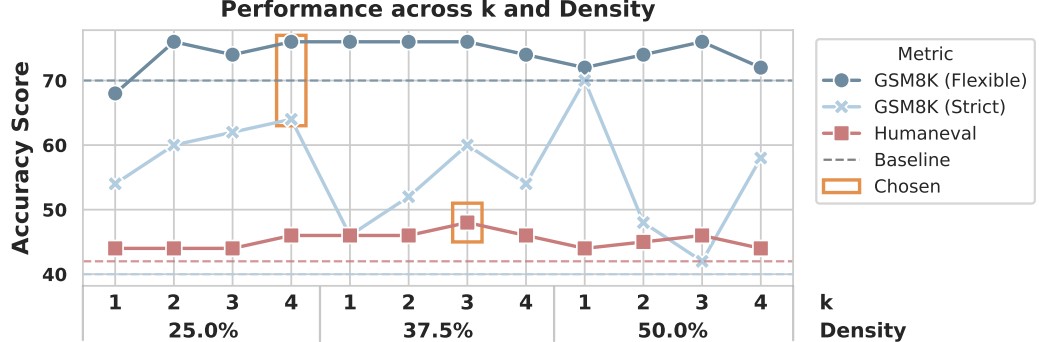

Figure 14: Ablation study on hyperparameters $k$ and $a$ for LLaDA-Instruct on subsets of GSM8K and HumanEval. The parameter $a$ is mapped to the retention density shown on the x-axis. Each dashed line represents the baseline performance for the solid line of the same color and metric.

The results of this search for LLaDA-Instruct are presented in Figure 14, conducted on 50-sample subsets of GSM8K and HumanEval. While using a small subset for tuning may introduce some variance, the findings provide clear directional insights. Our method with Gaussian Dropout consistently outperforms the baseline in nearly all configurations, with a single, minor exception for the GSM8K flexible-match accuracy score at $k = 1$ and a density of 25.0%. The suboptimal performance at $k = 1$ is expected, as at this value the Gaussian curve is relatively flat across the sampling window (as shown in Fig. 15), causing the sampling to degenerate into a near-uniform distribution.

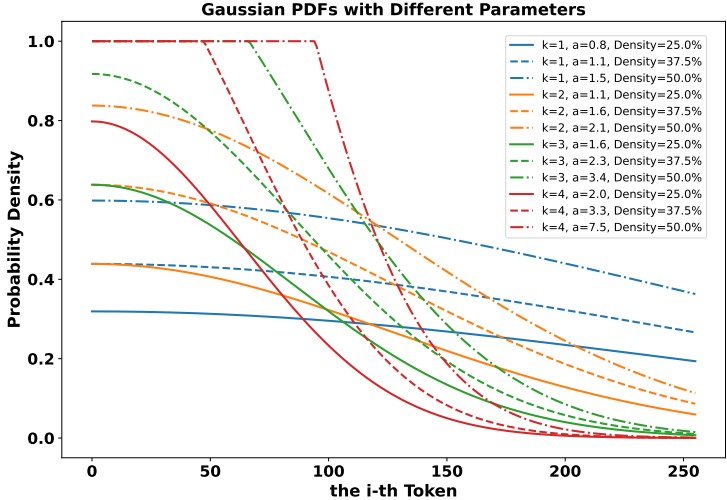

Figure 15: Different curvers of Gaussian pdfs with different parameters.

Focusing on the more effective range of $k \in [2, 4]$, we identify distinct optimal settings for each domain. For the mathematical reasoning task (GSM8K), a density of 25.0% provides the best balance of accuracy and efficiency. Although some settings with 50.0% density achieve a slightly higher strict-match score, they do so by preserving twice as many suffix tokens, which significantly undermines acceleration. We therefore select $k = 4.0$ and a density of 25.0% (from $a = 2.0$) as the optimal configuration. For code generation (HumanEval), a configuration of $k = 3.0$ and a density of 37.5% (from $a = 2.3$) yields the best performance.

Based on these findings, we adopt these hyperparameters for all subsequent math and code benchmarks for LLaDA-Instruct, with similar tuning for other models, as listed in Tbl. 9.

## D.2 Hyperparameter Settings

The primary hyperparameters for the Gaussian sampler are the decay rate factor $k$ and the magnitude scalar $a$, which controls the retention density. The optimal values used in our main experiments (Section 4.2) are summarized in Tbl. 9.

Table 9: The hyperparameters for Gaussian Sampler used in main experiments in Sec. 4.2

| Task | LLaDA-Instruct | | | | LLaDA-1.5 | | | | Dream-Base | | | |
|---|---|---|---|---|---|---|---|---|---|---|---|---|
| | $k$ | $a$ | Density | Window | $k$ | $a$ | Density | Window | $k$ | $a$ | Density | Window |
| GSM8K | 4.0 | 2.0 | 25.0% | 256 | 3.0 | 1.6 | 25.0% | 256 | 4.0 | 1.6 | 20.0% | 256 |
| Math | 4.0 | 2.0 | 25.0% | 256 | 3.0 | 1.6 | 25.0% | 256 | 4.0 | 1.6 | 20.0% | 128 |
| HumanEval | 3.0 | 2.3 | 37.5% | 512 | 3.0 | 1.6 | 25.0% | 128 | 3.0 | 2.3 | 37.5% | 128 |
| MBPP | 3.0 | 2.3 | 37.5% | 128 | 3.0 | 1.6 | 25.0% | 512 | 3.0 | 1.6 | 25.0% | 128 |

## D.3 Stochastic vs. Deterministic Suffix Dropout

To validate the design choice of using a stochastic (randomized) distance-decay strategy for *DPad*, we compared it against deterministic masking strategies. The goal was to determine if the model requires a dynamic "lottery ticket" search (as hypothesized in Sec. 3.3) or if a static selection of tokens would suffice.

We compared *DPad* against two deterministic baselines on LLaDA-1.5/GSM8K ($l_{max} = 256$):

- **Fixed Nearby:** Deterministically retains only the suffix tokens adjacent to the current block.
- **Fixed Distant:** Deterministically retains only the furthest suffix block.
- *DPad* **(Ours):** Dynamically resamples the dropout mask at each step.

Table 10: Stochastic vs. deterministic dropout on LLaDA-1.5/GSM8K.

| Method | Flexible | Strict |
|---|---|---|
| Baseline | 79.68 | 35.94 |
| Fixed Nearby | 71.65 | 55.19 |
| Fixed Distant | 28.51 | 21.83 |
| DPad | 79.98 | 68.84 |

**Results and Analysis.** As shown in Table 10, deterministic strategies significantly degrade performance. *Fixed Distant* causes a catastrophic collapse (Flexible accuracy drops to 28.51%), proving local context is non-negotiable. Crucially, *Fixed Nearby* also underperforms the baseline (71.65% vs 79.68%), indicating that a rigid cutoff creates permanent "blind spots" to necessary distant information. In contrast, *DPad* achieves the highest performance across both metrics. Its stochastic resampling allows the model to integrate information from the entire suffix range over time, effectively avoiding blind spots while preserving the noise-reduction benefits required for high strict-match accuracy.

## E Qualitative Case Studies

To provide qualitative insight into *DPad*'s impact, this section presents case studies illustrating its effect on in-context learning, format adherence, and generation conciseness.

### E.1 IMPROVED IN-CONTEXT LEARNING AND FORMAT ADHERENCE

A key benefit of *DPad* is its ability to improve strict-match accuracy by helping the model faithfully replicate the reasoning format from few-shot examples. Fig. 16 provides a clear example from the GSM8K benchmark. The baseline model produces the correct numerical answer but fails to follow the structured, step-by-step reasoning shown in the prompt. In contrast, *DPad* successfully mimics the desired format, leading to a successful strict match. This demonstrates that by pruning distracting, low-entropy suffix tokens, *DPad* allows the model to better focus on and learn from high-quality prompt examples.

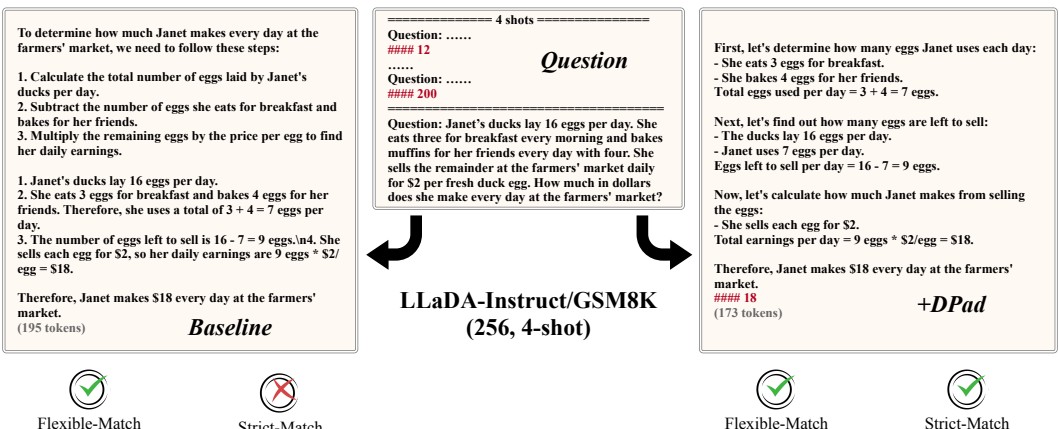

Figure 16: **Case study on format adherence from GSM8K.** This example contrasts the output of the baseline LLaDA-Instruct model with the *DPad*-enhanced version. The baseline provides the correct answer (Flexible-Match) but fails to follow the prompt's reasoning structure (failed Strict-Match). *DPad* generates both a correct and well-formatted response, passing both criteria.

### E.2 PROMOTING CONCISE AND EFFICIENT GENERATION

We also present a case study on the Dream model to illustrate how *DPad* influences generation behavior. Unlike the LLaDA models, Dream tends to exhaust its full token budget, resulting in verbose and inefficient outputs. *DPad* mitigates this by encouraging more concise and logically sound generations. Figure 17 shows that the baseline model generates a lengthy, 1021-token response that brute-forces a solution. *DPad*, however, produces a correct and elegant 69-token solution that terminates naturally.

## F DISCUSSION

### F.1 BEYOND A TRAINING-FREE METHOD: SUFFIX DROPOUT WITH SFT

While *DPad* demonstrates improved accuracy and computational efficiency across diverse benchmarks by strategically pruning the suffix space, we observe a performance degradation in very long-sequence generation, particularly at a context length of 2048 tokens. We attribute this behavior to a distributional shift introduced by suffix dropout. During pre-training, the model is optimized to predict token distributions conditioned on a full, continuous sequence of masked suffix tokens. In contrast, our distance-decay dropout forces the model at inference time to predict based on a much smaller, discontinuous set of masked suffix tokens. While moderate dropout helps the model focus on prompt semantics, such a large shift in the conditional distribution can hinder generalization, leading to degraded performance.

This training–inference distribution gap may be mitigated with supervised finetuning (SFT). In particular, the attention lottery hypothesis offers a useful perspective that can be incorporated into

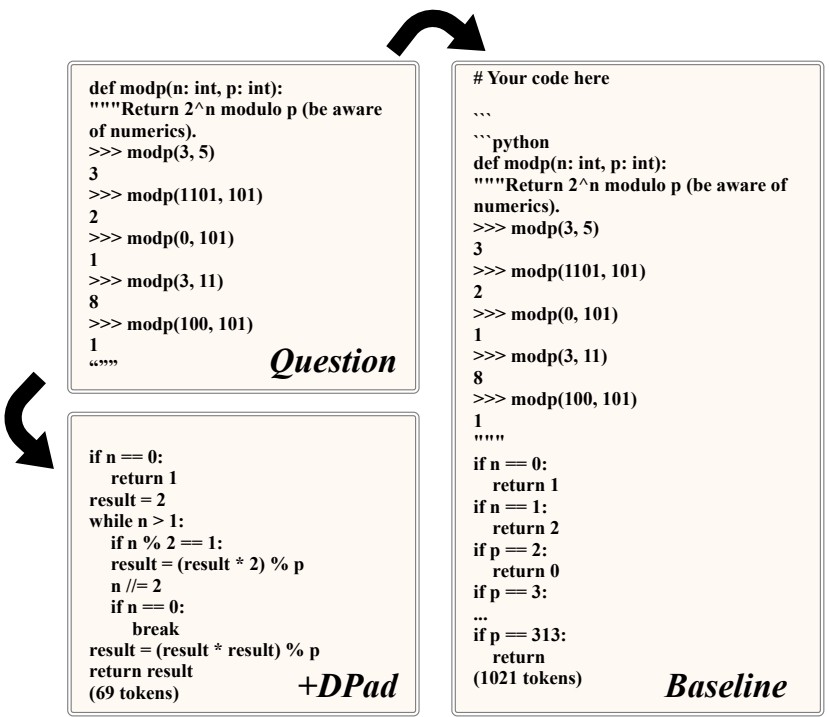

Figure 17: **Case study on generation conciseness from Dream/HumanEval.** A comparison of generated code for a modular arithmetic problem. The baseline model (1021 tokens) produces a highly verbose, unrolled solution to fill its token budget. In contrast, *DPad* (69 tokens) generates a compact, efficient, and logically superior solution, demonstrating improved reasoning.

training. We revise the learning objective as:

$$\mathcal{L}_{DPad}(\theta) = -\mathbb{E}_{x_0, t, M} \left[ \frac{1}{t} \sum_{i \in C} \log p_\theta \left( x_0^i \mid x_t^{(I_R \cup I_P \cup M)} \right) \right], \qquad (10)$$

where $M$ denotes a subset of masked suffix tokens sampled via distance-decay dropout at each training step, $I_R$ represents the prompt tokens, $I_p$ represents the generated prefix tokens, $\theta$ represents the model parameters, $t \in [0, 1]$ is degree of masking of samples from the forward masking process (Sec. 2.1), and the loss is computed over the current block tokens $C$. Using stochastic dropout masks rather than a fixed deterministic pattern enhances robustness, as the model does not overfit to any single dropout scheme. This revised objective explicitly integrates the notion of an attention lottery into training, encouraging the model to avoid wasting capacity by writing redundant information into distant suffix tokens that are likely to be pruned at inference.

Looking further ahead, one could even incorporate distance-decay dropout directly into the pre-training phase, allowing the model to learn sparsity from scratch. Such pre-training with sparse suffix attention would naturally align training and inference conditions, and may yield even stronger efficiency–accuracy trade-offs.

### F.2 COMPARISON TO SEMI-AUTOREGRESSIVE DIFFUSION AND BLOCK DIFFUSION

Block Diffusion models (Arriola et al., 2025) operate autoregressively at the block level, predicting each block conditioned only on its predecessors. As a result, their attention mechanism excludes access to subsequent blocks, in contrast to the directional attention employed by semi-autoregressive models.

While this design is computationally efficient, its strictly forward-looking nature introduces two major limitations. First, it is vulnerable to the reversal curse (Berglund et al., 2024), since it cannot capture long-range (beyond a single block) bidirectional dependencies that are crucial for tasks such as code generation, which often require iterative back-and-forth refinement (Gong et al.). Second, by discarding the suffix, it forfeits the ability to use suffix tokens as a *scratchpad* (see Sec. 3.1), thereby losing an important medium for contextual organization.

Our proposed distance-decay dropout interpolates between these paradigms. It retains suffix tokens to function as a scratchpad, while at the same time preserving the model's bidirectional learning ability, allowing information to be both written into and retrieved from the suffix scratchpad as needed.

## G  LLM USAGE DISCLOSURE

LLMs were used for (i) language editing and (ii) limited code assistance in code writing and debugging. All research code implementing our method and all experimental decisions were written and validated by the authors. We independently verified the correctness and reproducibility of the result.

