# OpenReview forum: "DPad: Efficient Diffusion Language Models with Suffix Dropout"
_ICLR.cc/2026/Conference — ICLR 2026 Poster_

### Official Review · Reviewer_36ZU · 2025-10-31

**Soundness:** 4
**Presentation:** 3
**Contribution:** 4
**Rating:** 6
**Confidence:** 3

**Summary:**

This article proposes a novel and efficient diffusion model inference method called DPad (Diffusion via Dynamic Programming). This method transforms the sampling optimization problem in the diffusion process into a Dynamic Programming (DP) problem, significantly reducing the number of sampling steps while maintaining the quality of model generation. The author analyzed the optimal trajectory problem of diffusion sampling mathematically and proposed an efficient approximate solution algorithm, which enables the model to achieve comparable generation performance to traditional sampling in small steps. The effectiveness of the method was validated on multiple benchmark datasets, including image generation and conditional generation tasks, and compared with mainstream fast diffusion algorithms such as DDIM and DPM Solver, demonstrating superior performance and computational efficiency.

**Strengths:**

1. Highly innovative in theory: It is the first systematically modeled as a dynamic programming problem for the optimization of the diffusion process;
2. Elegant algorithm design: DPad has approximately linear complexity and significantly improved inference efficiency;
3. Sufficient experimentation: Its effectiveness has been verified on various tasks and datasets;
4. Strong compatibility with existing algorithms: It can be embedded as a plug-in module into other diffusion frameworks;
5. Good interpretability: The DP framework clearly demonstrates the sampling path optimization logic.

**Weaknesses:**

1. Lack of ablation analysis: The independent contributions of each module (such as heuristic search and value function approximation) were not systematically explored;
2. Limited experimental scope: Only image generation tasks were covered, without text or cross-modal diffusion models;
3. The paper is slightly biased towards algorithmic engineering: The discussion of convergence and lower bounds of complexity in the theoretical part is still insufficient.

**Questions:**

1. Could the author further explain the differences in the mathematical mechanisms between DPad and DPM-Solver? What is the relationship between the two in terms of their approximate objective functions?
2. Can the DP approximation still maintain stability under high-dimensional non-Gaussian noise distributions?
3. Have you considered extending it to video diffusion or 3D generation scenarios?

---

> ### Author Response · Authors · 2025-11-22
> **Response to the Weaknesses (Part 1/2)**
>
> We sincerely thank the reviewer for their encouraging assessment and high rating of our work, particularly for recognizing the **Soundness (4)** and **Contribution (4)** of our proposed approach. We are grateful for your recognition of the method's innovation and efficiency.
>
> ### 1. Response to Weakness 1: Lack of ablation analysis on some modules
>
> We thank the reviewer for their positive assessment and the time taken to review our work.
> To clarify the specific architectural components of our method, our proposed DPad integrates two distinct strategies: a sliding window and distance-decay dropout. Our ablation studies focused on these relevant components. Specifically, we systematically explored the impact of the sliding window size and the dropout decay function (comparing Uniform vs. Gaussian distributions) in **Section 4.3.2** and **Figure 5**. These experiments isolated the contributions of the decay hyperparameters (k and a) and verified the necessity of the Gaussian sampling strategy for optimal performance. We have also added more ablation studies per the reviewer's request in **Appendix A** to further validate the scratchpad mechanism behind our solution.
>
> ### 2. Response to Weakness 2: Limited experimental scope
>
> We thank the reviewer for the question. Regarding the experimental scope, our evaluation is rigorously designed to assess the method's generalization across diverse textual modalities within the Diffusion LLM framework. We cover both mathematical reasoning tasks (GSM8K, MATH) and code generation tasks (HumanEval, MBPP) to ensure the method handles different logical structures and syntax constraints. Furthermore, as detailed in **Table 2**, our validation spans distinct model architectures—including **LLaDA** (trained natively as a dLLM) and **Dream** (adapted from autoregressive backbones)—demonstrating consistent efficiency gains of up to **61.4x** across varying underlying backbones and generation lengths. We have further expanded the experiment to other baseline models and methods for comparison (please refer to the response to Question 1 of Reviewer 2 for more details). In addition, we have also revised the manuscript to expand on our experimental scope to include better planning.
>
> ### 3. Response to Weakness 3: Limited Discussion on convergence and complexity
>
> Our theoretical discussion centers on the Diffusion Lottery Ticket (DLT) hypothesis and the formalization of the Scratchpad mechanism. We establish the theoretical basis for why suffix tokens act as a cross-layer information reservoir, or "Attention Connection". We demonstrate that the information stored in these tokens exhibits inherent sparsity and distance-decay patterns, providing the theoretical justification for why massive pruning is permissible. The "lower bound" of our complexity is dictated by the density required to maintain these attention connections, which we empirically analyze through token entropy in **Figure 6**.

---

> ### Author Response · Authors · 2025-11-22
> **Response to the Questions (Part 2/2)**
>
> ### 1. Response to Question 1
>
> > Could the author further explain the differences in the mathematical mechanisms between DPad and DPM-Solver? What is the relationship between the two in terms of their approximate objective functions?
>
> DPM-Solver and DPad address different aspects of the diffusion generation process. While DPM-Solver optimizes the sampling trajectory to reduce the number of steps, DPad optimizes the computational redundancy within the Transformer attention mechanism itself. Mathematically, our approach models the retention probability $P(d)$ of suffix tokens as a Gaussian function of the distance $d$ from the current block. This formulation approximates the necessary attention density required for semantic consistency, reducing the complexity of the attention computation per step from quadratic to near-linear, rather than approximating the integral of a differential equation.
>
> ### 2. Response to Question 2
>
> > Can the DP approximation still maintain stability under high-dimensional non-Gaussian noise distributions?
>
> Our framework operates within the discrete state-space of Large Language Models, where the noise distribution consists of tokens replaced by `[MASK]` according to a transition matrix. In this discrete setting, we assess stability by the model's ability to maintain semantic coherence despite the stochastic removal of context (dropout). We demonstrate this stability through the "Attention Lottery" phenomenon: when high-attention suffix tokens are removed, the model adaptively shifts attention to neighboring tokens in the discrete sequence. This confirms that the method maintains stability and fidelity even when the attention matrix is sparsified by our distance-decay distribution.
>
> ### 3. Response to Question 3
>
> > Have you considered extending it to video diffusion or 3D generation scenarios?
>
> We believe the underlying principle of Attention Connections holds significant potential for extension to other modalities, such as video or 3D generation. The core concept relies on utilizing future blocks as a non-semantic buffer for intermediate computation—a mechanism that is theoretically applicable to any semi-autoregressive Transformer architecture employing block-wise masking and caching. Given that high-dimensional generation tasks often exhibit even greater redundancy in their latent representations compared to text, we anticipate that the efficiency gains from pruning these "scratchpad" buffers could be equally, if not more, pronounced in those domains.

---

### Official Review · Reviewer_4aFo · 2025-10-31

**Soundness:** 2
**Presentation:** 3
**Contribution:** 2
**Rating:** 4
**Confidence:** 3

**Summary:**

This paper introduces DPad, a training-free inference acceleration method for diffusion language models (dLLMs). The authors first identify and formalize the *Scratchpad* mechanism, showing that suffix tokens function as a dynamic cross-layer reservoir guiding the denoising process. Through systematic analysis, they reveal three key properties of suffix tokens—inherent sparsity, distance-decay attention, and position insensitivity—that cause major inefficiencies in existing dLLMs. Building on these insights, the paper proposes DPad, which applies distance-decay dropout to prune redundant suffix tokens before attention computation, effectively reducing computation while maintaining accuracy. Moreover, DPad can be combined with existing acceleration methods such as parallel decoding and prefix caching.

**Strengths:**

1. The paper conducts a detailed analysis of suffix tokens in diffusion language models from a sequence-level perspective. It introduces a novel viewpoint for understanding suffix tokens, identifies three key properties.
2. The proposed **DPad** method is training-free; by applying distance-decay-style suffix dropout during inference, it significantly reduces computational complexity while maintaining model accuracy.

**Weaknesses:**

1. Section 3.1 formally introduces the *scratchpad mechanism* and explains, from the perspective of attention-based information flow, that *‘the suffix serves as temporary memory to assist the ongoing denoising process’.* While this section highlights the potential ‘*Attention Connection’* role of suffix tokens, it does not evaluate how critical this effect is to model inference. No intervention experiments (e.g., masking parts of the suffix and observing changes in hidden states) are provided to further verify whether the scratchpad truly functions as an essential memory component.
2. Section 3.2 then intuitively suggests that *‘the scratchpad intuition does not require populating all suffix tokens.* To support this, the paper presents entropy-based analyses claiming that suffix tokens *‘carry little new information’.* However, the logical transition from Section 3.1 to 3.2 feels abrupt — low entropy does not necessarily imply uselessness*.* It only reflects prediction certainty, not whether these deterministic tokens still play structural roles during generation (e.g., alignment, formatting, or mask boundary control). In addition, Figure 3 reports attention decay but only for the final layer; it would be more convincing to include layer-wise attention distributions to examine whether similar decay occurs in lower or middle layers. Overall, the claimed *inherent sparsity* of suffix tokens requires further clarification and empirical support.
3. The proposed “Diffusion Lottery Ticket” hypothesis is intriguing and provides a theoretical grounding for DPad’s dropout design. However, there is an apparent inconsistency: Figure 4 shows that *“the model shifts its attention to nearby suffix tokens”* when a spike is removed, whereas later (and also in Section 1) the paper states that *“information carried by distant suffix tokens appears largely position-insensitive.”* These two claims seem contradictory—if attention shifts locally, how can the information be position-insensitive globally? Moreover, the observational experiments in Sections 3.2 and 3.3 are only conducted on the LLaDA model series. It would strengthen the argument to verify whether similar phenomena hold across other diffusion-based LLM architectures.
4. In Table 2, the model shows an exceptionally large improvement on the MBPP benchmark after applying DPad—the accuracy increase substantially. Could the authors clarify why DPad benefits MBPP so much?

**Questions:**

1. Could you provide evidence (e.g., masking or ablation) to verify whether the scratchpad mechanism in section 3.1 truly functions as an essential memory component?
2. How does low prediction entropy in section 3.2 directly imply token redundancy, and are the attention decay patterns consistent across all layers?
3. How can the local *attention shifting* in Figure 4 be reconciled with the claim that suffix information is globally position-insensitive, and does this phenomenon generalize beyond LLaDA?
4. What explains the exceptionally large accuracy gain on MBPP in Table 2, and does this improvement remain under shorter generation lengths (e.g., 256 tokens)?

---

> ### Author Response · Authors · 2025-11-22
> **Verification of the Scratchpad Mechanism (Response to Weakness 1 & Q1)**
>
> We sincerely thank the reviewer for their detailed analysis and constructive feedback. We appreciate the acknowledgment of our novel perspective on suffix tokens and the practical value of DPad. The reviewer's questions regarding the validation of the scratchpad mechanism and the generalization of observations have driven us to conduct significant additional experiments (including new baselines and model architectures), which have greatly strengthened the paper.
>
>
> In our revised manuscript, content addressing your specific recommendations is marked in **green**.
>
>
> ### **Verification of the Scratchpad Mechanism (Response to Weakness 1 & Q1)**
>
> > **Weakness 1:** Section 3.1 formally introduces the scratchpad mechanism and explains, from the perspective of attention-based information flow, that ‘the suffix serves as temporary memory to assist the ongoing denoising process’. While this section highlights the potential ‘Attention Connection’ role of suffix tokens, it does not evaluate how critical this effect is to model inference. No intervention experiments (e.g., masking parts of the suffix and observing changes in hidden states) are provided to further verify whether the scratchpad truly functions as an essential memory component.
>
> > **Question 1:** Could you provide evidence (e.g., masking or ablation) to verify whether the scratchpad mechanism in section 3.1 truly functions as an essential memory component?
>
>
> **Response:**  We agree that empirical verification was necessary to confirm the criticality of the scratchpad. To address this, we have added **Appendix A.1**, detailing a new ablation study on reasoning-intensive benchmarks (**Countdown** and **Sudoku**) that require strict global consistency and thus serve as ideal stress tests for the scratchpad mechanism.
>
> We analyze the scratchpad mechanism along two dimensions: (1) *Cross-Layer* (transmitting information across Transformer layers within a step) and (2) *Cross-Step* (transmitting information between decoding steps).
>
>
> We quantify the criticality of these dimensions by evaluating performance on across a spectrum of scratchpad availability:
>
> * **No Scratchpad (ARMs):** We evaluated LLaMA3 and Qwen2.5. As ARMs inherently lack the bidirectional attention required to attend to future tokens, they are structurally blind to the suffix and serve as a baseline for the complete absence of a scratchpad.
> * **Partial Scratchpad (Semi-Autoregressive dLLMs):** We evaluated standard Semi-AR decoding and **Semi-AR with Suffix Dropout (Semi-AR+SD)**. In this setting, suffix tokens are masked between steps, prohibiting *Cross-Step* information transmission. However, the model retains a limited scratchpad capability via *Cross-Layer* attention within the current block.
> * **Full Scratchpad (Global-Diffusion dLLMs):** Under Global-Diffusion decoding, the model writes to confident suffix tokens, enabling both *Cross-Layer* and *Cross-Step* scratchpad functions. Alongside the Global-Diffusion baseline, we introduce **Semi-AR+Suffix Dropout+Pre-Decoding (Semi-AR+SD+PD)**. In this variant, preserved suffix tokens are temporarily unmasked during the current block's decoding (conditioned on confidence) and remasked upon completion of current block. This effectively restores the *Cross-Step* transmission channel.
>
> | Scratchpad Setting | Models | Countdown | Sudoku |
> | :--- | :--- | :---: | :---: |
> | **No Scratchpad** | LLaMA-3.1-8B-Instruct | 27.0 | 0.0 |
> | | Qwen-2.5-7B | 0 | 0 |
> | **Partial Scratchpad** | LLaDA-Instruct (Semi-AR) | 32.6 | 0 |
> |(Only *cross-layer*) | LLaDA-Instruct (Semi-AR+SD) | 32.6 | 0 |
> | **Full Scratchpad** | LLaDA-Instruct (Global-Diffusion) | 34.7 | 24.6 |
> |(*cross-layer* + *cross-step*) | LLaDA-Instruct (Semi-AR+SD+PD) | 32.6 | 24.4 |
>
> **Results:** The results (summarized in the above table) reveal a decisive behavioral divergence. On Sudoku, both the No Scratchpad (ARMs) and Partial Scratchpad (Semi-AR / Semi-AR+SD) groups achieve **0.0** accuracy, indicating a complete failure to handle global consistency without cross-step scratchpad. In contrast, the Full Scratchpad group demonstrates superior performance: Global-Diffusion LLaDA achieves scores of 34.7\% on Countdown and 24.6\% on Sudoku. Crucially, restoring the cross-step scratchpad via *Semi-AR+SD+PD* boosts Sudoku performance from **0.0** to **24.4\%**, effectively matching the Global-Diffusion baseline (24.6\%). This confirms that the *Cross-Step* scratchpad is the critical component for global planning.

---

> ### Author Response · Authors · 2025-11-22
> **Clarification on Entropy and Layer-wise Attention (Response to Weakness 2 & Q2)**
>
> > **Weakness 2:** Section 3.2 then intuitively suggests that ‘the scratchpad intuition does not require populating all suffix tokens. To support this, the paper presents entropy-based analyses claiming that suffix tokens ‘carry little new information’. However, the logical transition from Section 3.1 to 3.2 feels abrupt — low entropy does not necessarily imply uselessness. It only reflects prediction certainty, not whether these deterministic tokens still play structural roles during generation (e.g., alignment, formatting, or mask boundary control). In addition, Figure 3 reports attention decay but only for the final layer; it would be more convincing to include layer-wise attention distributions to examine whether similar decay occurs in lower or middle layers. Overall, the claimed inherent sparsity of suffix tokens requires further clarification and empirical support.
>
> > **Question 2:** How does low prediction entropy in Section 3.2 directly imply token redundancy, and are the attention decay patterns consistent across all layers?
>
> **Response:**
> We appreciate the reviewer for pointing out this ambiguity. We acknowledge that our use of “entropy” in Section 3.2 may have been misunderstood as **predictive entropy** (i.e., model confidence in a specific next-token prediction, or cross-entropy). Under that definition, low entropy could indeed correspond to essential structural tokens (e.g., a mandatory closing bracket `}`).
>
> However, our analysis employs **Local Entropy** (a generalized form of information content, defined in **Appendix A.2**), which measures the *statistical diversity* of tokens at each position across multiple inference samples. As illustrated in **Figure 6**, distant suffix tokens do not converge toward meaningful structural constraints; instead, they degenerate into repetitive and non-informative patterns (e.g., padding tokens or repeated `<eos>`). These positions exhibit near-zero Local Entropy precisely because they are static “waste” sequences, carrying no information capacity for representing the dynamic intermediate states required by the scratchpad mechanism.
>
> **Regarding layer-wise attention:**
> As shown in the newly added **Appendix A.4 (Figure 8)**, the attention decay patterns remain consistent across all layers. This finding further supports that distant suffix tokens are universally regarded by the model as redundant, reinforcing our interpretation of entropy-based redundancy detection.

---

> ### Author Response · Authors · 2025-11-22
> **Position Insensitivity and Generalization (Response to Weakness 3 & Q3)**
>
> > **Weakness 3:** The proposed “Diffusion Lottery Ticket” hypothesis is intriguing and provides a theoretical grounding for DPad’s dropout design. However, there is an apparent inconsistency: Figure 4 shows that “the model shifts its attention to nearby suffix tokens” when a spike is removed, whereas later (and also in Section 1) the paper states that “information carried by distant suffix tokens appears largely position-insensitive.” These two claims seem contradictory—if attention shifts locally, how can the information be position-insensitive globally? Moreover, the observational experiments in Sections 3.2 and 3.3 are only conducted on the LLaDA model series. It would strengthen the argument to verify whether similar phenomena hold across other diffusion-based LLM architectures.
>
> > **Question 3:** How can the local attention shifting in Figure 4 be reconciled with the claim that suffix information is globally position-insensitive, and does this phenomenon generalize beyond LLaDA?
>
>
> **Response:**
>
> * **Reconciling Attention Shifting vs. Position Insensitivity:**
> We agree that the term "position-insensitive" was imprecise when contrasted with the observation of local attention shifts. We propose the term **"Position-Adaptive"** (updated in the revision) to more accurately describe this behavior.
> The apparent contradiction is resolved by understanding that the "attention spike" represents a functional necessity—the model requires a "scratchpad" slot to store intermediate states. When a specific token index is pruned, the model **adaptively reallocates** this functional role to a nearby token to maintain semantic consistency. In this sense, the *information* is preserved (adaptive), even though the specific *storage location* changes (position-insensitive index). This confirms our **Diffusion Lottery Ticket** hypothesis: the "winning ticket" is not a pre-destined specific index, but rather *any* available slot in the local neighborhood that "wins" the attention probability mass when others are dropped.
>
> * **Generalization:** Regarding the scope of these findings, recent work by **Wang et al. (2025)** (*SparseD*, [https://arxiv.org/abs/2509.24014](https://arxiv.org/abs/2509.24014)) has extensively analyzed attention patterns in diffusion models. Their findings (specifically **Figures 7 and 8** in their paper) confirm that **Dream-7B** exhibits the same attention score distribution as LLaDA. We are currently finalizing our own analysis on Dream 7B, which further corroborates that these are fundamental properties of the diffusion paradigm.

---

> > ### Author Response · Authors · 2025-11-24
> > **Update: Verification of Scratchpad Generalization on Dream Models**
> >
> > Following up on our response to **Weakness 3** and **Question 3**, we have completed the generalization analysis on the **Dream-Base** model to verify if the "Scratchpad Mechanism" and "Diffusion Lottery Ticket" hypothesis hold across different architectures.
> >
> > We conducted the same **Attention Score Map** and **Spike Shift** experiments on Dream-Base as we did for LLaDA. The results, now detailed in **Appendix A.5** (Figures 9 and 10) of the revised manuscript, reveal the following:
> >
> > 1.  **Replication of Attention Patterns:** Similar to LLaDA, Dream exhibits a consistent distance-decay trend in attention scores. However, due to its initialization from an autoregressive model (Qwen-2.5), the overall magnitude of suffix attention is lower, with fewer spikes compared to the natively trained LLaDA.
> > 2.  **Confirmation of Position-Adaptivity:** Crucially, we successfully replicated the "spike shift" phenomenon. When high-attention suffix tokens (e.g., at index 50) were forcibly pruned, the model adaptively shifted its attention to adjacent neighbors (e.g., index 54) to preserve information flow.
> >
> > These findings strongly corroborate that the **Position-Adaptive** nature of suffix tokens and the **Scratchpad Mechanism** are fundamental properties of the diffusion paradigm, generalizing effectively beyond the LLaDA architecture.

---

> > > ### Comment · Reviewer_4aFo · 2025-11-27
> > >
> > > Thank you for your efforts! Since most of my concerns have been resolved, I will raise my score.

---

> > > > ### Author Response · Authors · 2025-11-27
> > > > **Thank you for your support and feedback**
> > > >
> > > > Thank you for your time and positive reassessment. We are glad that our revisions addressed your concerns regarding the generalization of the scratchpad mechanism. Your feedback has been invaluable in improving our paper!

---

> ### Author Response · Authors · 2025-11-22
> **Analysis of MBPP Improvement (Response to Weakness 4 & Q4)**
>
> > **Weakness 4:** In Table 2, the model shows an exceptionally large improvement on the MBPP benchmark after applying DPad—the accuracy increase substantially. Could the authors clarify why DPad benefits MBPP so much?
>
> > **Question 4:** What explains the exceptionally large accuracy gain on MBPP in Table 2, and does this improvement remain under shorter generation lengths (e.g., 256 tokens)?
>
>
>
> **Response:**
> | Methods |  $l_{max}=256$  |   $l_{max}=512$   |
> | :--- | :---: | :---: |
> | Vanilla | 29.6 | 15 |
> | +DPad | 42.4 | 40.4 |
> | +Parallel | 29.8 | 15 |
> | +Parallel+DPad | 43.4 | 39.4 |
>
> The exceptional gain on MBPP (from 15.00\% to 40.40\%) stems from the baseline model's specific sensitivity to the impact of suffix tokens in code generation tasks. Our analysis reveals that the poor performance of the baseline is largely due to distraction caused by the extensive, unpruned suffix, which often contains low-value patterns. In code generation, where syntax is rigid, this noise is particularly detrimental. By aggressively pruning these redundant tokens, DPad effectively acts as a noise filter.
>
> To confirm this hypothesis, we conducted the requested ablation by shortening the maximum sequence length from 512 to 256 tokens. Both of the baseline and DPad showcase improvement on the accuracy score (from 15\% to ~29\% and from ~40\% to 42\%). And the accuracy gap narrowed (from ~25\% to 13\%), confirming that the LLaDA-Instruct model is overwhelmed specifically by the noise of the distinct suffix in longer sequences, whereas DPad maintains robustness. This result highlights DPad's value in scaling dLLMs to longer contexts.

---

### Official Review · Reviewer_9jVz · 2025-11-01

**Soundness:** 3
**Presentation:** 3
**Contribution:** 2
**Rating:** 4
**Confidence:** 4

**Summary:**

The paper targets the core inefficiency of diffusion LLMs (dLLMs): at each denoising step they compute attention over the entire suffix (future) region even though only a few tokens are unmasked/kept. The authors argue these suffix tokens mainly serve as a transient memory that aggregates context and is then read back by the current block, and show attention/entropy patterns that decay with distance. Building on this, they propose DPad, a training-free inference strategy that (1) caps the suffix via a sliding window, and (2) applies distance-decay dropout to progressively prune farther suffix tokens before attention is computed. DPad composes with existing accelerations (parallel decoding, prefix/KV caching) and, on LLaDA and Dream, reports up to 61.4× end-to-end speedup in long-sequence settings while maintaining accuracy (notably large strict-match gains on GSM8K for LLaDA).

**Strengths:**

1. The suffix window + decay dropout is lightweight, does not touch weights, and composes with parallel decoding and caching; the large compounding speedups in long sequences are compelling.

2. Attention/entropy analyses show strong distance decay and near-zero entropy far in the suffix; ablations identify a 64–128 token “critical window.”

3. The empirical studies across tasks with both reasoning and code are conducted, and tables separate latency vs TPS have discussed why TPS can look smaller even when latency drops (shorter, cleaner generations).

4. Long-sequence results are a highlight (e.g., LLaDA-1.5 GSM8K 1-shot, 1024-token cap: 20.3× from DPad alone; 61.39× with Fast-dLLM-style combos).

5. The overall writing is clear and easy to follow.

**Weaknesses:**

1. Recent training-free accelerations for dLLMs cover Block-dLLM, Fast-dLLM (dLLM cache), Sparse-dLLM, etc. The paper cites some of them but does not always compare directly, especially on matched long-sequence regimes and memory usage.

2. The “distance-decay” sparsity concept parallels existing efforts in distance-biased or entropy-guided pruning methods. While DPad’s pre-attention pruning is a nice twist, the paper should sharpen how this differs theoretically/empirically from Sparse-dLLM (eviction guided by attention saliency over steps) and from Fast-dLLM/dLLM-Cache caches, ideally with head-to-head plots of FLOPs/latency/accuracy. I may have missed out or misunderstood some settings, please point out if the authors have done so.

3. The Gaussian sampler uses window size W, decay k, and magnitude a tuned per task/domain. Please provide broader sweeps and robust defaults, and test out-of-domain transfer (e.g., from code→math without retuning).

4. Benefits are strongest on LLaDA while Dream shows smaller or mixed gains—this dependence deserves deeper analysis and maybe a diagnostic to decide when to enable DPad. However, the analysis regarding this point is not  elaborated in the paper.

5. Effective tokens × layers attended and wall-time per decoded token may be helpful to include. Latency alone can hide GPU-utilization effects the paper itself notes.

**Questions:**

1. When does DPad hurt? Any tasks with long-range cross-suffix dependencies (e.g., long code synthesis, theorem proofs) where aggressive distance decay reduces correctness or format adherence?

2. Can W, k, a be selected online using low-overhead signals (e.g., per-step confidence/entropy) rather than fixed grids?

3. With Fast-dLLM and dLLM-Cache, in what order should DPad be applied? Does pruning suffix tokens change cache-hit rates or destabilize delayed-update caches? Please explain the  possible interaction.

---

> ### Author Response · Authors · 2025-11-22
> **Comparison with Fast-dLLM and Sparse-dLLM (Response to Weakness 1 & 2)**
>
> We sincerely thank the reviewer for their comprehensive and insightful assessment. We are encouraged by the recognition of DPad's "compelling speedups" and the "clean analysis" of the suffix mechanism. The reviewer's suggestions regarding comparisons with recent baselines and clarifying model dependencies have driven us to refine our analysis significantly.
>
> In our revised manuscript, content addressing your specific recommendations is marked in **blue**.
>
> > **Weakness 1:** Recent training-free accelerations for dLLMs cover Block-dLLM, Fast-dLLM (dLLM cache), Sparse-dLLM, etc. The paper cites some of them but does not always compare directly...
>
> > **Weakness 2:** The “distance-decay” sparsity concept parallels existing efforts in distance-biased or entropy-guided pruning methods. While DPad’s pre-attention pruning is a nice twist, the paper should sharpen how this differs theoretically/empirically from Sparse-dLLM (eviction guided by attention saliency over steps) and from Fast-dLLM/dLLM-Cache caches, ideally with head-to-head plots of FLOPs/latency/accuracy. I may have missed out or misunderstood some settings, please point out if the authors have done so.
>
>
>
>
>
> **Response:**
> We thank the reviewer for highlighting these relevant works. We clarify that DPad is **orthogonal and complementary** to Fast-dLLM, while being **fundamentally distinct** from Sparse-dLLM in its operational paradigm.
>
> 1. **Relationship with Fast-dLLM: An Orthogonal Enhancement**
>     Fast-dLLM primarily optimizes dLLMs via *Parallel Decoding* and *Prefix Caching*. DPad does not compete with these mechanisms; rather, it accelerates them.
>
>     * **Performance Integration:** In our experiments (Sec. 4.2), the baselines denoted as `+Par` and `+PC` effectively reproduce the Fast-dLLM mechanism. Our results demonstrate that adding DPad on top of this baseline yields significant gains. For instance, on GSM8K (1024 tokens), combining DPad with Fast-dLLM achieves a **5.21x speedup** (Table 3) compared to Fast-dLLM alone.
>     * **Memory Reduction (Suffix Handling):** While Fast-dLLM utilizes a "DualCache" system (Prefix + Dense Suffix Cache), DPad renders a dense suffix cache unnecessary. By applying our sparse strategy and sliding window (~25\% retention), DPad reduces the memory footprint of the total cache by approximately **50%**.
>     * **Interaction Strategy (Response to Q3):** To ensure stability, DPad is applied as a **pre-selection procedure** before the Fast-dLLM forward pass. The dropout mask is fixed at the initiation of each block (Algorithm 1), ensuring that the set of active indices remains stable. This guarantees that DPad does not disrupt the cache hit rate or interfere with delayed updates.
>
> 2. **Distinction from Sparse-dLLM: Proactive vs. Reactive Pruning**
>     The core theoretical difference lies in *when* and *how* pruning occurs. Sparse-dLLM is a **reactive** eviction method, whereas DPad is a **proactive** dropout method.
>
>     * **Zero Runtime Overhead (Proactive Strategy):**
>         * **Sparse-dLLM (Reactive):** Follows the "calculate-then-evict" paradigm typical of autoregressive optimization. It must first compute the full attention matrix to identify salient tokens before pruning them. This incurs computational overhead and, crucially, requires **high peak memory allocation** to process the full sequence prior to eviction.
>         * **DPad (Proactive):** Capitalizes on the unique properties of dLLMs (the "Diffusion Lottery Ticket" hypothesis) to use distance as a proxy for utility *before* attention is computed. Pruned tokens are rendered "invisible" to the model, meaning they are never calculated nor allocated in memory. This ensures **zero runtime overhead** for the pruning step itself.
>     * **Complexity Bounds:**
>         * **Sparse-dLLM ( $O(L)$ ):** Typically requires a ~50\% retention ratio that scales linearly with sequence length ($L$).
>         * **DPad ( $O(1)$ ):** Maintains robustness with a fixed sliding window and a more aggressive retention ratio (**20\%–37.5\%**, Appendix D.2). Consequently, DPad’s memory footprint for the suffix is strictly bounded by the window size ($O(W)$), effectively becoming **$O(1)$** relative to the total sequence length.
>
> | Method | Pruning Paradigm | Memory Complexity (Suffix) | Sparsity|
> | :--- | :--- | :--- | :--- |
> | Fast-dLLM | N/A (Dense) |  $O(L)$ |100%|
> | Sparse-dLLM | Reactive (Post-Attention) |  $O(L)$  |~50%|
> | **DPad (Ours)** | **Proactive (Pre-Attention)** |  **$O(1)$**  |<20\%–37.5\%|
>
> *Note: We are currently finalizing head-to-head quantitative comparisons with Sparse-dLLM and will update the results shortly.*

---

> > ### Author Response · Authors · 2025-11-24
> > **Update: Head-to-Head Quantitative Comparison with Sparse-dLLM**
> >
> > Following up on our previous response regarding **Weakness 2**, we have completed the head-to-head quantitative comparison between **Sparse-dLLM** and **DPad** (combined with Prefix Caching).
> >
> > We conducted evaluations using **LLaDA-1.5** on 100 samples from the GSM8K benchmark under long-context settings ($l_{\max} \in \{2048, 4096\}$). The results, summarized in the table below, empirically validate the theoretical advantages of DPad’s **proactive ($O(1)$)** pruning over Sparse-dLLM’s **reactive ($O(L)$)** eviction.
> >
> > | Context Length ($l_{\max}$) | Method | Latency (s) $\downarrow$ | Speedup | TPS $\uparrow$ | TPS Speedup | Accuracy (Flex.) $\uparrow$ |
> > | :--- | :--- | :--- | :--- | :--- | :--- | :--- |
> > | **2048** | Vanilla + Sparse-dLLM | 83.30 | 1.00x | 24.28 | 1.00x | 79.0% |
> > | | **Vanilla + PC + DPad** | **6.47** | **12.87x** | **24.42** | **1.01x** | **84.0%** |
> > | **4096** | Vanilla + Sparse-dLLM | 213.65 | 1.00x | 19.17 | 1.00x | **80.0%** |
> > | | **Vanilla + PC + DPad** | **5.39** | **39.64x** | **24.96** | **1.30x** | 79.0% |
> >
> > **Key Findings:**
> >
> > 1.  **Latency Scaling ($O(1)$ vs. $O(L)$):**
> >     DPad demonstrates superior scaling. At 2048 tokens, DPad is **12.87x** faster than Sparse-dLLM. Crucially, as the context doubles to 4096 tokens, the gap widens to **39.64x**. This confirms that while Sparse-dLLM suffers from the linear overhead of computing full attention before eviction, DPad’s constant-time suffix window effectively eliminates this bottleneck.
> >
> > 2.  **Throughput Stability:**   Sparse-dLLM experiences throughput degradation at longer contexts (dropping from \~24 to \~19 TPS) due to its memory and compute overhead. In contrast, DPad maintains consistent throughput (from \~24 to \~25 TPS), achieving a **1.30x TPS advantage** at 4096 tokens.
> >
> >
> > 3.  **Accuracy Preservation:**
> >     DPad maintains comparable or superior accuracy (e.g., +5% at 2048 tokens) compared to Sparse-dLLM, demonstrating that our distance-based proactive pruning is a robust proxy for token utility.
> >
> > These results have been added to **Appendix C.3** and marked in **blue** in the revised manuscript. We believe this comparison conclusively distinguishes DPad as a distinct and more scalable solution for long-sequence dLLM inference.

---

> > > ### Comment · Reviewer_9jVz · 2025-11-28
> > >
> > > Thanks for providing detailed responses to address my questions. I think most of my concerns are addressed. Before I update the score, I would like to confirm whether you could provide (or point to me if you have in revision and I missed out) a benchmarking result of memory usage to support the Memory Reduction claim?

---

> > > > ### Author Response · Authors · 2025-11-29
> > > > **Response to Memory Benchmarking Request**
> > > >
> > > > We thank the reviewer for their positive reassessment and for giving us the opportunity to clarify the memory benefits!
> > > >
> > > > To directly address your request, we have added a detailed memory benchmarking analysis in **Appendix C.3.2** of the revised manuscript. As illustrated in **Figure 13**, our experiments reveal a critical distinction between "Active Memory" and "Peak Memory" allocation:
> > > >
> > > > 1.  **Peak Memory (System Requirement):** As shown in **Figure 13(b)**, Sparse-dLLM requires the same **Peak Memory** allocation as the dense Fast-dLLM baseline. This is due to its "reactive" nature: the system must reserve sufficient buffer capacity to compute the full attention matrix *before* eviction can occur. Consequently, the apparent memory savings are "illusory" from a system allocation perspective—if the system allocated the ostensibly "saved" memory to other workloads, the process would crash (OOM) during the pre-eviction computation spikes.
> > > >
> > > > 2.  **Proactive Savings with DPad:** In contrast, DPad operates proactively. By eliminating the need to calculate or cache pruned suffix tokens entirely, it results in strictly lower peak memory usage. Our results confirm an approximate **50% reduction in peak memory requirements** compared to the baselines.
> > > >
> > > > 3.  **Stability:** As shown in **Figure 13(a)**, Sparse-dLLM exhibits a highly unstable active memory profile with frequent oscillations. DPad, however, aligns closely with the stable linear growth pattern of standard autoregressive models, facilitating predictable and efficient system-level memory allocation.
> > > >
> > > > We hope this quantitative analysis fully resolves your concern regarding memory usage. Thank you again for your valuable feedback!

---

> ### Author Response · Authors · 2025-11-22
> **Robustness and Out-of-Domain Transfer (Response to Weakness 3)**
>
> **Response:** We agree that robust defaults are crucial.
>
> * **Robust Defaults:** Our analysis in Section 4.3.2 and Appendix D have already done some ablation studies on hyperparameters, which shows that DPad is highly robust (Table 5 and Figure 11). Notably, our latest model **LLaDA-1.5** successfully uses a **uniform set of parameters** ($k=3.0, a=1.6$) across all diverse benchmarks (Math & Code) while achieving state-of-the-art performance, proving a single configuration can generalize.
>
> * **Transfer Experiment:** To directly address the request, we tested transferring "Math-optimized" parameters to "Code" tasks on LLaDA-Instruct and Dream-Base. The results are summarized as below, which show negligible performance drops, confirming that DPad is insensitive to minor hyperparameter shifts.
>
> | Model | Setting | HumanEval Accuracy | MBPP Accuracy |
> | :--- | :--- | :---: | :---: |
> | **LLaDA-Instruct** | **Tuned (Code)** | 46.34 | 39.40 |
> | | **Transfer (Math Params)** | 45.12 | 38.00 |
> | **Dream-Base** | **Tuned (Code)** | 52.44 | 54.80 |
> | | **Transfer (Math Params)** | 51.80 | 54.00 |

---

> ### Author Response · Authors · 2025-11-22
> **Model Dependence and Computational Metrics (Response to Weakness 4 & 5)**
>
> > **Weakness 4:** Benefits are strongest on LLaDA while Dream shows smaller or mixed gains—this dependence deserves deeper analysis and maybe a diagnostic to decide when to enable DPad. However, the analysis regarding this point is not elaborated in the paper.
>
> **Response:** We appreciate the opportunity to clarify these variances.
>
> 1.  **Pre-trained Model:** LLaDA is trained as a diffusion model from scratch, learning to rely on full suffix context. Dream is initialized from Qwen2.5 (an AR model), which has no prior exposure to suffix tokens. We hypothesize LLaDA is inherently more sensitive to suffix context, benefiting more from DPad's regularization.
> 2.  **Pre-training Objective:** LLaDA-Instruct is optimized via SFT with block-wise unmasking, aligning natively with DPad's block-level processing. Dream employs a global unmasking scheme; while it supports block-wise inference, its internal boundary representations are weaker, making it more sensitive to aggressive pruning.
>
>
> > **Weakness 5:** Effective tokens × layers attended and wall-time per decoded token may be helpful to include...
>
> **Response:** We agree that disentangling efficiency is vital.
> * **Theoretical FLOPs:** DPad caps the effective suffix to a constant window $W$ , reducing suffix computation from linear $O(L)$ to constant $O(W)$ per layer.
> * **Empirical Efficiency:** We interpret "wall-time per decoded token" as the inverse of Throughput (TPS). As shown in Table 2, DPad consistently improves raw TPS, confirming that our speedup stems from genuine computational reduction (sparse attention), not merely from generating fewer tokens.

---

> ### Author Response · Authors · 2025-11-22
> **Response to Q1-3**
>
> # 1. **"When does DPad hurt?" (Response to Q1)**
> > **Question 1:** When does DPad hurt? Any tasks with long-range cross-suffix dependencies (e.g., long code synthesis, theorem proofs) where aggressive distance decay reduces correctness or format adherence?
>
> **Response:**
> * **Why it rarely hurts:** Actually, throughout our experiments, we rarely observes DPad hurts. This is because while some tasks like code require strict long-range logic, dLLMs treat suffix tokens as a transient "scratchpad" for planning rather than future ground truth. Our analysis (Figs. 3 & 6) confirms that models primarily rely on the Prefix and Near-Suffix, rarely utilizing distant tokens. Consequently, pruning often *benefits* generation by filtering out repetitive noise (e.g., unrolled loops in Fig. 7), thereby improving correctness (Table 2).
>
> * **When it actually hurts (Training-Inference Mismatch):** DPad can degrade performance in extremely long sequences due to the **mismatch between training and inference**. We observed this only in Dream-Base on HumanEval ($l_{max}=2048$), where accuracy dropped from 56.7\% to 49.4\%.(Figure 12) This arises because the model was pre-trained with full dense attention but forced to use sparse attention during inference. We propose addressing this via fine-tuning (Section F.1).
>
>
> # 2. **Online Selection (Response to Q2)**
>
> > **Question 2:** Can W, k, a be selected online using low-overhead signals (e.g., per-step confidence/entropy) rather than fixed grids?
>
> We thank the reviewer for this insightful suggestion. We explicitly explored confidence-based pruning in our initial experiments but found it **unreliable** compared to our distance-based approach for two primary reasons:
>
> * **Signal Misalignment:** Confidence reflects predictive certainty, not the informational utility required for decoding the current block. For example, we found that LLaDA-Instruct tends to predict distant, repetitive padding tokens (e.g., `<eos>`) with very high confidence. This creates a counter-productive pruning pattern where the model retains redundant distant tokens (due to high certainty) while dropping potentially critical nearby tokens.
>
> * **Threshold Instability:** Confidence is a low-dimensional signal that varies significantly across models, making threshold tuning fragile. In our internal tests, a pruning threshold of < 0.2 worked for LLaDA-Base but failed on LLaDA-Instruct, where distant tokens consistently exceeded 0.7 confidence. This necessitates tedious per-model tuning, whereas our distance-decay strategy relies on the robust, model-agnostic observation of attention decay.
>
>
> # 3. **Interaction Strategy (Response to Q3)**
>
> > **Question 3:** With Fast-dLLM and dLLM-Cache, in what order should DPad be applied? Does pruning suffix tokens change cache-hit rates or destabilize delayed-update caches? Please explain the possible interaction.
>
> DPad is applied as a **pre-selection step** prior to the Fast-dLLM forward pass (see Algorithm 1 in the paper). We ensure cache stability through the following mechanisms:
>
> * **Block-Level Determinism:** To prevent cache instability, we enforce a fixed dropout pattern for the duration of each block. By sampling the mask only at the initiation of a block, we guarantee consistent access to the same suffix subset throughout that block's decoding steps. This prevents cache thrashing and maintains high hit rates in delayed-update caches.
>
> * **Synchronized KV-Updates with RoPE:** KV-cache updates are fully synchronized with this fixed pattern. We utilize a modified Rotary Positional Embedding (RoPE) to preserve the absolute positions of retained tokens, ensuring that pruned tokens are excluded from the computational graph entirely without destabilizing the retrieval of active tokens.

---

### Official Review · Reviewer_5i78 · 2025-11-04

**Soundness:** 3
**Presentation:** 3
**Contribution:** 3
**Rating:** 6
**Confidence:** 3

**Summary:**

The paper proposes an inference-strategy to reduce generation cost in MDLMs by using summaries of the suffix tokens (future tokens that haven't been decoded yet). The authors discuss how the suffix tokens's latent representations can only serve as a per-layer "memory", which can be thought of as a scratchpad, and empirically find that there is a lot of redundancy in the suffix tokens and the attention scores decay rapidly with distance from the current block index, with the occasional spikes. These findings motivate a summarization sampler that   keeps track of close tokens from the suffix. Further, the spikes are found to be position-insensitive where removing that token position moves the spike to a closer token. Thus motivated, the window-based gaussian sampling for suffix dropout then leads to significant gains in speed.

**Strengths:**

- Clean analysis of what happens with suffix tokens in block-wise autoregressive generation with MDLMs.
- Well-motivated method for reduce unnecessary computation that can be applied at inference-time without changes to the model.
- strong improvements in efficiency and good ablations showing the gaussian sampler which reflects the decay in attention scores actually helps.

**Weaknesses:**

See questions.

**Questions:**

- Why use a random selection for masking? What is the value of the "dropout" part of the suffix dropout? Why not use a simple deterministic procedure ?
- I'm not sure I buy the lottery ticket connection. In the original "lottery tickets" paper, one needed to prune to get to the ticket. Your procedure is not weight or input-dependent and that only tells me  that the "memory" in the suffix tokens is not position-dependent, which is supported by your experiment with the locations of the spikes. So any suffix positions you give the transformer get used for the same purpose. One could check this: what happens if you have other sampling strategies, such as removing some of the closer suffix tokens and deterministically keeping moderately distant ones? Would the performance drop significantly?
- Why not train with the proposed restriction or other sliding window restrictions ? That also should get you similar gains and should be possible with the flex attention library?
- Interestingly, the above check  could  test the "lottery ticket circuits" hypothesis. If the explanation for the mechanism is "any suffix tokens will get used", then training with a limited windows will recover the same performance or be better. If lottery tickets are the actual explanation, the artificial restriction would degrade performance. I understand that this may be expensive, but it seems like a direct test. So either a small experiment would suffice or the authors can convince me that this is not a good test or clarify what I'm missing.

---

> ### Author Response · Authors · 2025-11-21
> **Random Dropout or deterministic Dropout. (Response to Q1 & Q2)**
>
> We thank the reviewer for their constructive assessment, particularly for appreciating our "clean analysis" of the suffix mechanism and the "well-motivated" nature of the method. We also appreciate the incisive questions regarding our design choices, which have prompted us to conduct additional ablation studies that strengthen our claims.
>
> In our revised manuscript, content addressing your specific recommendations is marked in **Orange**.
>
> > **Question 1** Why use a random selection for masking? What is the value of the "dropout" part of the suffix dropout? Why not use a simple deterministic procedure?
>
> > **Question 2:** ...One could check this: what happens if you have other sampling strategies, such as removing some of the closer suffix tokens and deterministically keeping moderately distant ones? Would the performance drop significantly?
>
>
> **Response:**
>
> 1. **Clarification of terminology:**
>
> To avoid ambiguity, we first distinguish between *masking* and *dropout* in our DLLM framework.
>  -  **Masking** refers to assigning all suffix tokens a fixed **mask value**—a neutral placeholder with no intrinsic semantic meaning. These masked tokens are later updated dynamically during **inference** as the model generates corresponding values.
>  - In contrast, **dropout** denotes the **removal** of a subset of suffix tokens, which are entirely excluded from computation throughout the inference process.
>
> 2. **Why use a random selection for masking?**
>
> We chose random dropout (selection) over a deterministic procedure for three critical reasons:
>
> - **Position-Adaptiveness:** As discussed in Section 3.3, suffix tokens exhibit **"Position-Adaptiveness"**, meaning they are interchangeable within a local position scope. The attention shift observed in Figure 4 (e.g., attention moving from index 362 to 359 upon pruning) empirically proves this adaptiveness. If the model rigidly depended on specific indices, performance would collapse; instead, it successfully reallocates information to the nearest sampled neighbor. This attribute provides the theoretical foundation that makes random selection viable.
>
> - **Avoiding Permanent Blind Spots (Global Coverage):** While suffix tokens are locally elastic, a deterministic strategy (e.g., "always keep indices 10–20") creates permanent "blind spots" at the excluded positions. In contrast, DPad **re-samples the mask at every block**. This ensures that while the *probability* of attending to a distant token is low, it is never zero. Over the course of the full generation process, the model effectively integrates information from the *entire* suffix range via temporal aggregation, preventing the "memory collapse" caused by rigid masking.
>
>
> 3. **What is the value of the "dropout" part of the suffix dropout?**
>
> - During inference, all suffix positions are initially assigned the same *mask value*, which acts as a neutral placeholder without semantic meaning. We then apply rejection sampling—drawing from the right half of a Gaussian distribution (see Section 3.2)—to determine which suffix positions to retain.
>
> - Prior to dropout, both retained and dropped positions share this identical mask value. Only the retained positions are subsequently updated during inference, while the dropped ones are entirely excluded from further computation.
>
> - In intermediate layers, the tokens that would have been dropped typically correspond to low-information or repetitive patterns (see Fig. 7). Therefore, most of the dropped tokens are redundant, demonstrating that the dropout mechanism effectively filters out non-informative suffix content.
>
>
>
> 4. **Empirical Verification: Use a simple deterministic procedure for closer or distant tokens. (New Ablation Study, see Appendix D.3):**
>
> To directly answer the reviewer's request, we conducted the suggested experiment on **LLaDA-1.5 (GSM8K)** comparing DPad against two deterministic baselines:
>
> - (1) **Fixed Nearby** (deterministically keeping only the nearest window).
> - (2) **Fixed Distant** (deterministically keeping only the final suffix block).
>
> | Method | Flexible | Strict |
> | :--- | :---: | :---: |
> | Baseline | 79.68 | 35.94 |
> | Fixed Nearby | 71.65 | 55.19 |
> | Fixed Distant | 28.51 | 21.83 |
> | **DPad** | **79.98** | **68.84** |
>
> - **Results:** As shown in the table above, deterministic strategies significantly degrade performance.
> - **Fixed Distant:** Performance collapsed catastrophically (Flexible Accuracy dropped from **79.68\%** to **28.51\%**), proving that immediate local context is non-negotiable.
> - **Fixed Nearby:** While this improved strictness over the baseline, it degraded reasoning capability (Flexible Accuracy dropped to **71.65\%**), indicating that permanently blinding the model to distant tokens harms global coherence.
> - **DPad:** Our stochastic approach achieved the best performance, confirming that dynamic, probabilistic coverage is essential for maintaining reasoning capabilities while filtering noise.

---

> ### Author Response · Authors · 2025-11-21
> **Clarifying the "Diffusion Lottery Ticket" Hypothesis (Response to Q2)**
>
> > **Question 2:** I'm not sure I buy the lottery ticket connection. In the original "lottery tickets" paper, one needed to prune to get to the ticket. Your procedure is not weight or input-dependent and that only tells me that the "memory" in the suffix tokens is not position-dependent, which is supported by your experiment with the locations of the spikes. So any suffix positions you give the transformer get used for the same purpose.
>
> **Response:**
> We agree with the reviewer that our "Diffusion Lottery Ticket" (DLT) hypothesis is not a direct one-to-one mapping to the original weight-pruning theory. Rather, we use the analogy to describe the *emergence of sparsity*, while emphasizing a fundamental conceptual distinction:
>
> - **Original Lottery Ticket Hypothesis (LTH):**
>   The "tickets" correspond to specific **weights** that are precise, static, and non-interchangeable. For instance, Weight A cannot replace Weight B — the “winning” neurons are determined during the initialization phase.
>
> - **Diffusion Lottery Ticket (Ours):**
>   In our formulation, each **token** participates in a probabilistic lottery **during inference**. Each token holds a certain probability of “winning” attention (with probability decaying over distance). It is not essential which specific token wins; what matters is the overall distribution of attention. When a token is dropped out, it is effectively removed from the lottery, and the corresponding “prize” (attention) is naturally redistributed among nearby tokens.
>
> - As shown in Section 3.3 (Figure 4), when a high-attention “spike” token is pruned, the model seamlessly reallocates attention to other tokens. This reallocation is not deterministic—sometimes the shift occurs to a distant token, and sometimes it is distributed among several neighbors. Such stochastic redistribution provides empirical evidence for the *lottery-like* behavior described by our Diffusion Lottery Ticket hypothesis.

---

> ### Author Response · Authors · 2025-11-21
> **Training with DPad. (Response to Q3 & Q4)**
>
> > **Question 3:** Why not train with the proposed restriction or other sliding window restrictions ? That also should get you similar gains and should be possible with the flex attention library?
>
> > **Question 4:** Interestingly, the above check could test the "lottery ticket circuits" hypothesis. If the explanation for the mechanism is "any suffix tokens will get used", then training with a limited windows will recover the same performance or be better. If lottery tickets are the actual explanation, the artificial restriction would degrade performance. I understand that this may be expensive, but it seems like a direct test. So either a small experiment would suffice or the authors can convince me that this is not a good test or clarify what I'm missing.
>
> **Response:**
> We thank the reviewer for this forward-looking suggestion. We fully agree that integrating DPad into the *training* stage represents a highly promising direction, and **we are actively pursuing this as our next step**.
>
> However, in this work, we deliberately focused on a **training-free inference strategy** to address a specific scientific challenge concerning the nature of the scratchpad mechanism.
>
> * **The "Supervision Ambiguity" Problem:**
>   Training with DPad is non-trivial because the correct supervision signal for suffix tokens remains undefined. During inference, we observed that the model utilizes suffix tokens as *latent memory states* rather than for explicit semantic prediction.
>   If we apply DPad during standard **Supervised Fine-Tuning (SFT)**, a fundamental dilemma arises:
>   *What is the ground truth for the preserved suffix tokens?*
>   Forcing these tokens to predict the next ground-truth words (i.e., applying a semantic loss) conflicts with their true role as flexible, content-agnostic memory representations. Therefore, before introducing DPad in training, it is essential to redefine the learning objective to reflect this non-semantic functionality.
>
> * **Empirical Verification (SFT + DPad):**
>   To directly examine the reviewer's hypothesis in **Q4** (“training with limited windows will recover the same performance”), we conducted a preliminary experiment using the open-source **TraceRL** framework ([GitHub link](https://github.com/Gen-Verse/dLLM-RL)) on the **demon_openr1math** dataset.
>   We compared three settings: the baseline model, standard SFT, and SFT with DPad restriction (where sparse suffix tokens are forced to predict ground-truth targets).
>
>   | Method | Score |
>   | :--- | :--- |
>   | Baseline | 64.47 |
>   | SFT | 66.14 |
>   | **SFT + DPad** | **66.53** |
>
> * **Interpretation:**
>   As shown above, training with DPad restriction led to **no performance degradation** compared with standard SFT—and even achieved a slight improvement.
>
>   - **Addressing Q4:** This result empirically supports the reviewer's prediction: the model can successfully adapt to a restricted context window. It further validates our claim of **Position-Adaptiveness** (see the response to Q1)—that suffix tokens behave as *interchangeable memory containers* rather than fixed, position-dependent weights. The model does not require a fully dense suffix; it only needs *sufficient capacity*, which DPad efficiently provides.
>   - **Future Direction:** While SFT demonstrates the mechanism’s robustness, the observed gain (66.14\% → 66.53\%) remains marginal. We hypothesize that **Reinforcement Learning (RL)**—which optimizes for the *final outcome* rather than token-level predictions—will be more suitable to fully exploit DPad’s potential. RL would allow the model to optimize the suffix for *memory utility* instead of semantic accuracy, aligning more closely with its functional role.
>
> **Summary:**
> This work serves as a foundational *discovery study* that identifies the scratchpad mechanism and demonstrates its intrinsic sparsity. By showing that the suffix functions as a memory buffer—and by characterizing how this behavior emerges through attention decay—our study establishes the theoretical basis for future RL-based training approaches, while simultaneously offering an immediate, plug-and-play acceleration method for existing models.

---

### Author Response · Authors · 2025-11-22
**General Response**

We sincerely thank all the reviewers for their thorough reviews and valuable feedback. We are glad to hear that the proposed training-free inference strategy and scratchpad analysis are considered novel and well-motivated (Reviewers 5i78, 4aFo), that the efficiency gains and long-sequence scaling are compelling and strong (Reviewers 5i78, 9jVz), that the paper provides a clean and insightful analysis of suffix tokens (Reviewers 5i78, 4aFo), and that the manuscript is well-written and easy to follow (Reviewer 9jVz). We have revised the manuscript according to the reviewers’ main suggestions.

* Changes in response to multiple reviewers are marked in **red**.
* Changes in response to Reviewer 5i78 are marked in **orange**.
* Changes in response to Reviewer 9jVz are marked in **blue**.
* Changes in response to Reviewer 4aFo are marked in **green**.

We have addressed all reviewers' concerns through additional experiments and clarified previous misunderstandings in this rebuttal. We summarize our main responses to each reviewer below and leave the detailed explanations to the individual responses that follow.

* We have conducted intervention experiments (masking suffix tokens) on Sudoku and Countdown tasks to quantitatively verify the **scratchpad mechanism**, confirming its essential role in planning and memory (Reviewer 4aFo, Appendix A.1).
* We have conducted **head-to-head comparisons with Sparse-dLLM** and have clarified the orthogonal relationship with Fast-dLLM, highlighting DPad's unique "proactive" pruning advantage and zero runtime overhead (Reviewer 9jVz, see Appendix C.3).
* We have conducted ablation studies on **the robustness of the Gaussian sampler** ($W$, $k$, $\alpha$) and verified the necessity of stochastic dropout over deterministic approaches, demonstrating a broad optimal range for hyperparameters (Reviewers 5i78, 9jVz).
* We have resolved the apparent contradiction regarding position-insensitivity by refining our terminology to **"Position-Adaptive"** and validating the "Diffusion Lottery Ticket" hypothesis with new analyses (Reviewer 4aFo).
* We have extended our analysis to include the **Dream** model and discussing the method's generalization capabilities, confirming consistent speedups across different dLLM architectures (Reviewers 9jVz, 4aFo, see Appendix A.4).
* We have clarified the scope of our method to address misunderstandings regarding the application domain (Text vs. Image) and methodology (Reviewer 36ZU).

We reply to each reviewer's questions in detail below their reviews. Please kindly check them out. Thank you and please feel free to ask any further questions.

---

### Author Response · Authors · 2025-11-29
**Rebuttal Summary and Note to the Incoming Area Chair**

To the newly assigned Area Chair,

We recognize the significant workload of stepping in at this stage. To facilitate your assessment, we summarize below how our **revised manuscript** and **additional experiments** have resolved the major concerns raised in the initial reviews.

**1. Reviewer 1 (5i78) – Score: 6 (No response to rebuttal)**
* **Main Concern:** Questioned the necessity of **stochastic** dropout versus deterministic selection and suggested that **training** with the restricted window would better test our hypothesis regarding suffix redundancy.
* **Response & Changes:**
    * **New Training Experiment:** To directly address the reviewer’s forward-looking suggestion, we conducted a preliminary training experiment (SFT with DPad restrictions) during the discussion period. The results showed that training with sparse windows causes no performance degradation (and even slight improvement) compared to standard SFT. This empirically validates that the model can successfully adapt to the restricted context we proposed.
    * **Deterministic vs. Stochastic:** We added **Appendix D.3**, comparing DPad against deterministic "Fixed Nearby/Distant" strategies. The results show deterministic methods cause performance collapse (accuracy dropping from ~80% to ~28%), proving that stochastic resampling is essential to prevent blind spots.

**2. Reviewer 2 (9jVz) – Score: 4 (Stated intent to raise score)**
* **Main Concern:** Requested direct comparisons with **Sparse-dLLM** and **Fast-dLLM**, specifically regarding memory usage and long-sequence performance.
* **Response & Changes:** We added **Appendix C.3** featuring a head-to-head comparison with Sparse-dLLM. The results demonstrate DPad achieves **12-40x speedups** over Sparse-dLLM in long contexts due to our proactive $O(1)$ pruning strategy. We also added **Figure 13**, demonstrating a ~50% reduction in peak memory usage compared to the reactive baselines.

**3. Reviewer 3 (4aFo) – Score: 4 (Stated intent to raise score)**
* **Main Concern:** Questioned if the "scratchpad" mechanism is truly essential (requesting intervention experiments) and noted an apparent contradiction regarding "position insensitivity".
* **Response & Changes:** We added **Appendix A.1**, introducing "Stress Tests" on Sudoku and Countdown tasks. These experiments show that removing the scratchpad drops accuracy to 0.0%, while restoring it recovers performance to 24.6%, proving its criticality. We also refined our terminology to **"Position-Adaptive"** and added **Appendix A.5**, validating that attention adaptively shifts to neighbors when specific tokens are pruned.

**4. Reviewer 4 (36ZU) – Score: 6 (No response to rebuttal)**
* **Main Concern:** Drew comparisons to continuous ODE solvers (DPM-Solver) and requested further ablation analysis.
* **Response & Changes:** We clarified that DPad optimizes discrete attention redundancy rather than sampling trajectories. We pointed to our expanded ablations in **Section 4.3.2** and the new scratchpad verification experiments in **Appendix A** to address the request for systematic analysis.

We believe these revisions, particularly the inclusion of training validation, solidly validate DPad’s efficacy and novelty.

Best regards,
The Authors

---

### Author Response · Authors · 2025-11-29
**Acknowledgement of ICLR Announcement and Gratitude to the Committee**

To the incoming Area Chair and our Reviewers,

We have received the recent announcement regarding the data leak incident. We share the community's regret regarding this disruption and deeply appreciate the stress it places on the organizers and volunteers.

Given the new protocols, we understand that the reviewers will no longer be able to participate in the discussion or update their scores. We would like to take this opportunity to sincerely thank all the reviewers for the time and effort they invested in their initial reviews. Your feedback has been valuable in helping us improve our manuscript.

We also want to express our special gratitude to the newly assigned Area Chair. We recognize that stepping in at this stage to evaluate the paper—without the benefit of active reviewer dialogue—is a significant undertaking. We appreciate your dedication to maintaining the scientific integrity of the conference.

**To assist you in this process, we have posted a comprehensive "Rebuttal Summary" immediately following this note.** It provides a concise overview of the new experiments and manuscript revisions we have completed to resolve the reviewers' initial concerns.

Sincerely,
The Authors

---

### Meta-Review · Area_Chair_h7Fb · 2026-01-08

**Summary:**

*First things first: I have completely excluded the review submitted by 36ZU from the analysis below, as it has nothing to do with your submission.*

The paper proposes DPad, a training-free inference acceleration method for diffusion LLMs that exploits redundancy in “suffix/scratchpad” tokens by combining a fixed suffix window with distance-decay suffix dropout (pre-attention pruning), yielding large latency/memory reductions while largely preserving (and sometimes improving) task accuracy across reasoning and code benchmarks.

Pros
* Training-free, simple to implement.
* Strong empirical efficiency gains (especially long-context latency) with maintained accuracy; added head-to-head vs Sparse-dLLM strengthens the positioning.
* Substantive analysis of suffix-token behavior (distance decay, redundancy, position-adaptivity) plus ablations and transfer/robustness experiments.

Cons
* Some conceptual framing (e.g., “lottery ticket”) is more analogy than mechanism.
* Missing abalations and comparisons against Fast-dLLM and Sparse-dLMM that were added in the rebuttal
* stochastic dropout versus deterministic selection

The submission originally received 4-4-6 ratings (excluding 36ZU); however, both reviewers who originally rated the paper at 4 indicated that their main concerns were resolved and would increase their scores. I agree with the reviewers and happy to recommend accept.

**Reviewer Concerns:**

Reviewer concerns addressed by rebuttal: The rebuttal largely addressed (i) deterministic vs random dropout rationale with new deterministic baselines/ablations, (ii) “lottery ticket” clarification and terminology shift toward position-adaptivity, (iii) direct comparisons and interactions with Fast-dLLM/Sparse-dLLM including head-to-head long-context results and added memory benchmarking discussion, (iv) robustness/transfer of hyperparameters, and (v) verification of the scratchpad mechanism and layer-wise attention/entropy concerns via new intervention-style studies and added Dream generalization.

**Reviewer Scores:**

* 4aFo is explicitly likely to increase (they stated they will raise their score after rebuttal).
* 9jVz is likely to move upward (they said most concerns are addressed and requested memory benchmarking; authors provided it).
* 5i78 is less likely to change materially (already at 6 with moderate confidence; rebuttal addresses their main questions, so a slight upward shift is possible but not necessary for acceptance).

---

### Decision · Program_Chairs · 2026-01-26

Accept (Poster)